# Regional responses of surface ozone in Europe to the location of high-latitude blocks and subtropical ridges

Carlos Ordóñez[1], David Barriopedro[1,2], Ricardo García-Herrera[1,2], Pedro M. Sousa[3], Jordan L. Schnell[4,5]

[1] Departamento de Física de la Tierra II, Facultad de Ciencias Físicas, Universidad Complutense de Madrid, 28040 Madrid, Spain

[2] Instituto de Geociencias (IGEO, CSIC-UCM), Madrid, Spain

[3] Instituto Dom Luiz, Faculdade de Ciências, Universidade de Lisboa, 1749‑016 Lisboa, Portugal

[4] Department of Earth System Science, University of California, Irvine, CA, USA

[5] Program in Atmospheric and Oceanic Sciences, Princeton University, Princeton, NJ, 08540, USA

*Correspondence to*: Carlos Ordóñez (carlordo@ucm.es)

**Abstract.** This paper analyses for the first time the impact of high-latitude blocks and subtropical ridges on near-surface ozone in Europe during a 15-year period. For this purpose, a catalogue of blocks and ridges over the Euro-Atlantic region is used together with a gridded dataset of maximum daily 8-hour running average ozone (MDA8 $O_3$) covering the period 1998–2012. The response of ozone to the location of blocks and ridges with centres in three longitudinal sectors (Atlantic, ATL, 30º–0º W; European, EUR, 0º–30º E; Russian, RUS, 30º–60º E) is examined. The impact of blocks on ozone is regionally and seasonally dependent. In particular, blocks within the EUR sector yield positive ozone anomalies of ~5–10 ppb over large parts of central Europe in spring and northern Europe in summer. Over 20 % and 30 % of the days with blocks in that sector register exceedances of the 90[th] percentile of the seasonal ozone distribution at many European locations during spring and summer, respectively. The impacts of ridges during those seasons are subtle and more sensitive to their specific location, although they can trigger ozone anomalies above 10 ppb in northern Italy and the surrounding countries in summer, eventually exceeding European air quality targets. During winter, surface ozone in the northwest of Europe presents completely opposite responses to blocks and ridges. The anticyclonic circulation associated with winter EUR blocking, and to a lesser extent with ATL blocking, yields negative ozone anomalies between -5 ppb and -10 ppb over the UK, northern France and the Benelux. Conversely, the enhanced zonal flow around 50˚–60˚ N during the occurrence of ATL ridges favours the arrival of background air masses from the Atlantic and the ventilation of the boundary layer, producing positive ozone anomalies of ~5 ppb in an area spanning from the British Isles to the northern half of Germany. We also show that multiple linear models on the seasonal frequency of occurrence of these synoptic patterns can explain a considerable fraction of the interannual variability in some winter and summer ozone statistics (mean levels and number of exceedances of the 90[th] percentile) over some regions of western Europe. Thus, this work provides the first quantitative assessments of the

remarkable but distinct impacts that the anticyclonic circulation and the diversion of the zonal flow associated with blocks and ridges exert on surface ozone in Europe. The findings reported here can be exploited in the future to evaluate the modelled responses of ozone to circulation changes within chemical transport models (CTMs) and chemistry-climate models (CCMs).

## 1 Introduction

Air pollution results from a combination of elevated emissions and unfavourable weather conditions. One of the air pollutants of most concern for public health is surface ozone ($O_3$). Ozone in the troposphere is formed by photochemical reactions involving non-methane volatile organic compounds (NMVOCs), carbon monoxide (CO) and nitrogen oxides ($NO_x{\equiv}NO+NO_2$). The oxidation of methane ($CH_4$) also contributes to tropospheric ozone formation at the global scale. Another source of tropospheric ozone is the injection of $O_3$ from the stratosphere, while dry deposition to the Earth's surface and chemical loss provide the main sinks. Ozone is known to be detrimental not only to humans' health (e.g. Brunekreef and Holgate, 2002) but also to ecosystems, reducing plant primary productivity and crop yields (e.g. Ashmore, 2005). Moreover, tropospheric $O_3$ is a greenhouse gas. The past increase in tropospheric $O_3$ is estimated to have contributed 0.40 (0.20 to 0.60, 5 to 95% confidence interval) W $m^{-2}$ to direct radiative forcing (RF) since the pre-industrial era (Myhre et al., 2013). There is also robust evidence that tropospheric ozone can reduce the natural uptake of carbon dioxide ($CO_2$) by decreasing plant productivity (Sitch et al., 2007), although quantitative estimates of the associated indirect RF are still uncertain (Myhre et al., 2013).

A large number of observational studies have established statistical relationships between near-surface ozone concentrations and meteorological variables at the local and regional scales for over two decades (e.g. Bloomfield et al., 1996; Tarasova and Karpetchko, 2003; Ordóñez et al., 2005; Camalier et al., 2007; Seo et al., 2014). Comprehensive summaries can be found in the reviews by Jacob and Winner (2009) and Fiore et al. (2012, 2015). While ozone is favoured by elevated temperatures and regional stagnation in summer (e.g. Jacob and Winner, 2009), stable weather conditions generally accelerate the loss of ozone close to the surface in winter through the enhanced effect of dry deposition and reaction with nitrogen monoxide (NO) to form nitrogen dioxide ($NO_2$), a process known as ozone titration (e.g. Laurila et al., 1999; Ordóñez et al., 2005).

Some studies have addressed the connection of tropospheric ozone with circulation patterns, but the few analyses available for Europe are often restricted to specific regions. As an example, some studies have established objective links between circulation weather types (CWTs) and surface ozone in western Iberia (Carvalho et al., 2010; Russo et al. 2014) as well as in northwest and central Europe (e.g. Demuzere et al., 2009, 2011). Furthermore, past research efforts have traditionally been focused on the export of pollution from North America and the injection of stratospheric ozone into the troposphere (see e.g. Monks et al., 2009, 2015, and references therein), as well as on the impact of the North Atlantic Oscillation (NAO) regimes (Hurrell, 1995; Jones et al., 1997). Creilson et al. (2003) suggested that during positive phases of the NAO, characterized by stronger than usual Azores high and Icelandic low, the enhanced westerlies across the North Atlantic lead to increased

springtime ozone in western Europe as a consequence of the long-range transport of pollution plumes from North America. More recently, Pausata et al. (2012) showed that the commonly used NAO index (NAOI) is able to capture the link existing between atmospheric dynamics and European surface ozone concentrations in winter and spring, while a modified NAO index is needed to detect the atmospheric circulation - ozone relationship in summer.

Monthly teleconnection indices associated with large-scale circulation patterns can explain intra- and inter-annual fluctuations of the intensity of the zonal flow, but such one-dimensional monthly-based approaches limit our understanding of the complex spatial patterns occurring at intra-monthly time scales (e.g. Sousa et al., 2106a). Therefore, the day-to-day variability of the $O_3$ concentrations at the regional scale may be better captured by analysing daily circulation indices or the occurrence, persistence and position of specific synoptic-scale weather patterns. For example, some attention has already
been paid to the role of frontal passages associated with mid-latitude cyclones as well as to the relative position and strength of meteorological systems in Europe (e.g. Ordóñez et al., 2005; Doche et al., 2014). During summer the first days after a frontal passage are usually accompanied by an increase in temperature and stagnation of the air masses, leading to elevated ozone concentrations over central Europe, while the opposite effect can be observed in winter as the stability favours the ozone loss by titration and dry deposition (Ordóñez et al., 2005). The relative position and strength of the Azores anticyclone
and Middle Eastern depression (Doche et al., 2014), as well as the subsidence of ozone-rich air masses from the upper troposphere and lower stratosphere (e.g. Doche et al., 2014; Zanis et al., 2014), also influence the variability of lower tropospheric ozone over the Mediterranean in summer. Many studies have highlighted the complexity of the ozone dynamics in the Mediterranean region, where the interaction between the synoptic scale and mesoscale circulations needs to be considered. A number of sub-synoptic processes such as land-sea breeze and channelling effects due to terrain features
which favour the re-circulation of air masses and the formation of ozone reservoir layers, vertical mixing along the sloped terrain, and subsidence are known to be relevant in the region (e.g. Millán et al., 2000; Gangoiti et al., 2001; Cros et al., 2004; Drobinski et al., 2007; Flocas et al., 2009; Velchev et al., 2011; Kalabokas, 2013). This study will be focused only on the role of the synoptic scale.

Comprehensive analyses of the impact of synoptic patterns on surface ozone in Europe, considering their seasonal effects
and temporal scales larger than those of episodic cases, are not common. Among the synoptic features which are expected to impact surface ozone and other air pollutants one can mention the frequency and persistence of high pressure systems with an anticyclonic circulation such as high-latitude blocks and subtropical ridges. These systems are often associated with stable weather conditions in all seasons and above normal temperatures in summer under the anticyclonic centre, thus setting favourable conditions for the formation of ozone in that season. There are conceptual differences in the definition of
blocking and subtropical ridge patterns although sometimes they induce similar anomalies in the surface climate (e.g. Barriopedro et al., 2010b; Sousa et al., 2016a). Atmospheric blocking is associated with persistent, slow-moving high pressure systems that interrupt the prevailing westerly winds of middle and high latitudes, and therefore the eastward progress of extratropical storm systems (Christensen et al., 2013). Changes in the frequency and duration of blocking events have a significant impact on temperature and precipitation in winter (Fraedrich et al., 1993; Trigo et al., 2004; Masato et al.,

2012; Sousa et al., 2016a). Atmospheric blocking has also been associated with temperature extremes in Europe, including heat waves (Matsueda, 2011; Katsafados et al., 2014) and cold spells (Buehler et al., 2011; Cattiaux et al., 2010), presenting stronger links with collocated warm temperature extremes than with collocated cold extremes over mid to high latitude land regions (Pfahl and Wernli, 2012). On the other hand, subtropical ridges are low latitude structures characterized by positive geopotential height anomalies extending from sub-tropical latitudes towards extra-tropical regions such as southern Europe (e.g. Sousa et al., 2016b) and by northward displacements of the Atlantic eddy-driven jet stream (Woollings et al., 2011). The occurrence of strong and persistent mid-latitude anticyclonic ridges over the Eastern North Atlantic plays a key role in triggering severe droughts in western Iberia during winter (Santos et al., 2009a, 2009b). Moreover, Santos et al. (2015) found that temperature extremes in Iberia during summer 2003 were associated with enhanced anticyclonic circulation over northern Africa and the western-central Mediterranean, extending as a ridge towards northern France and diverting the westerly flow to northern Europe. García-Herrera et al. (2010) also showed that the atmospheric conditions during the 2003 "mega-heatwave", which triggered strong air pollution episodes over large areas of western Europe, did not display the typical signatures of canonical blocking patterns. Finally, recent analyses have documented the distinctive regional responses of European temperatures to both blocking and subtropical ridges (Sousa et al., 2016b). However, the impact of these synoptic patterns on surface air quality (AQ) on multiannual time scales has not been assessed yet.

In this study we conduct the first comprehensive analysis of the seasonal impacts of high-latitude blocks and subtropical ridges on near-surface ozone. A recent 15-year gridded dataset of daily ozone mixing ratios over Europe is used for this purpose. Section 2 describes the ozone and meteorological data used as well as the methodology applied to build the catalogue of high-latitude blocks and subtropical ridges. The impact of blocks and ridges on ozone is separately examined for the seasons with the highest levels of this pollutant (spring and summer) in Sect. 3 and for winter in Sect. 4. Section 5 investigates whether the occurrence of these synoptic patterns can control the interannual variability of ozone. Finally, Sect. 6 summarises and discusses the main results, while Sect. 7 outlines some of the implications of our findings.

## 2 Data and methods

### 2.1 Ozone and meteorological data

We use an interpolated dataset of observed maximum daily 8-hour running average near-surface ozone (MDA8 $O_3$) over Europe during the period 1998–2012. These gridded data have been created with the objective mapping algorithm of Schnell et al. (2014, 2015), who compiled and merged hourly observations from the European Monitoring and Evaluation Programme (EMEP) and the European Environment Agency's air quality database (AirBase) to calculate hourly surface ozone averaged over 1° x 1° grid cells. This product has been initially used by Schnell et al. (2014, 2015) to evaluate global chemical transport models (CTMs) and chemistry–climate models (CCMs). More recently, this dataset has been proved to be useful for the examination of the influence of atmospheric circulation and meteorological conditions on extreme ozone concentrations in Europe (Otero et al., 2016). Overall, seasonal linear trends of MDA8 $O_3$ are small for most regions during

the period of analysis. The main exception is the Balkans, which presents significantly positive ozone trends at the 5 % level and will be omitted from the regional analyses presented here.

Daily geopotential heights at 500 hPa (Z500) from the NCEP/NCAR meteorological reanalysis dataset at 2.5°×2.5° horizontal resolution (Kalnay et al., 1996) are employed for the detection of high-latitude blocks and subtropical ridges (see
Sect. 2.2). Daily mean sea level pressure (MSLP) and 850 hPa horizontal wind fields from this reanalysis dataset are also used to compare the low level flow patterns associated with ridges and blocks, and assess their impacts on MDA8 $O_3$. Additional meteorological fields such as daily mean total cloud cover, daily mean downward shortwave and longwave radiation fluxes at surface, and temperature at 2 m (daily mean and daily maximum) have been analysed; we will not show results for these specific fields but they will be discussed when their impact is relevant.

Before introducing the methodology used for the detection of blocks and ridges, we will briefly discuss some of the main climatological features found in the 15-year MDA8 $O_3$ dataset. This will help interpret the impacts of these synoptic patterns on ozone in the following sections. In this paper seasons are defined as follows: spring (March, April, May; MAM), summer (June, July, August; JJA), autumn (September, October, November; SON) and winter (December, January, February; DJF). For the sake of brevity, autumn will be omitted from our analyses, as $O_3$ mixing rations are not particularly high in this
season and the features that blocks and ridges imprint on surface ozone are weaker than those in the other seasons. The reason for this is that during autumn there is a strong transition, with ozone responses to these synoptic patterns in September similar but smaller in magnitude than those in summer, while in November such responses resemble those of winter, leading to an overall compensation of positive and negative ozone anomalies.

Figure 1 illustrates the seasonal climatology of MDA8 $O_3$ (shaded areas), MSLP (black contour lines) and 850 hPa winds
(arrows) in spring (top), summer (middle) and winter (bottom) during the 1998–2012 period. On average over the European continent, surface MDA8 $O_3$ mixing ratios are somewhat higher in spring (~41 ppb) than in summer (~39 ppb). This is not surprising as a distinct spring maximum has been observed at many locations of the northern hemisphere. The spring ozone maximum is likely a result from the combination of dynamical/transport processes and photochemistry at relatively unpolluted locations (Monks, 2000), while polluted sites exhibit a double peak or a wide spring-summer maximum, with the
summer maximum being more pronounced at daytime and hence attributed to photochemical processes (Monks et al., 2015). Observational studies have also shown that the Mediterranean region presents a marked summertime ozone maximum (e.g. Richards et al., 2013, and references therein). Actually, in summer there is a strong north-south gradient, with low mixing ratios in the UK and most of Scandinavia and high in southern Europe, including the areas around the Mediterranean coast (Figure 1, middle). During this season, ozone levels in northern (central-southern) Europe are lower (higher) than those
registered in spring. The highest MDA8 $O_3$ mixing ratios in summer (average seasonal values above 55 ppb) are found for the Po Basin, where very high levels of ozone and other photooxidants both at the surface and within the boundary layer have been reported in previous field experiments (e.g. Martilli et al., 2002; Neftel et al., 2002; Thielmann et al., 2002; Steinbacher et al., 2005; Liu et al., 2007). The picture in winter is completely different as the European continent is predominantly a sink for ozone during this season (Laurila, 1999). Cold temperatures, low radiation and short days in winter

inhibit ozone formation, while shallow mixing layers favour the accumulation of pollutants close to the surface and therefore the depletion of surface ozone by reaction with NO as well as dry deposition. The bottom panel of Figure 1 shows the low winter ozone levels over most of the continent, in particular in regions of elevated emissions such as the Benelux and the Po Basin. Winter surface ozone presents the highest levels over the northern and western coasts as these are directly under the influence of maritime air masses into the European continent advecting background ozone, which has undergone relatively low loss by dry deposition and photochemical processes. This is favoured by the zonal westerly flow prevailing over northern Europe, as can be inferred from the MSLP and wind fields in the figure.

## 2.2 Climatology of high-latitude blocks and subtropical ridges

We use the same catalogue of high-latitude blocks and subtropical ridges as Sousa et al. (2016a, 2016b). As mentioned above, the NCEP/NCAR meteorological reanalysis dataset at 2.5°×2.5° has been used to create the catalogue. This horizontal resolution is coarser than that of other reanalysis datasets, but it is appropriate to characterise large-scale phenomena such as blocks and ridges. Blocks are detected by applying a simplified version of the method described by Barriopedro et al. (2006), where they are defined as large-scale reversals of the Z500 meridional gradient and must fulfil some conditions on area overlap during consecutive days and minimum duration. For this study we have considered that reversals must have a minimum longitudinal extension of 12.5° during their whole lifetime and a minimum duration of 5 days. To identify ridges and avoid their overlap with blocks, the detection algorithm sets some north/south boxes (black squares in Figure 2) whose latitudinal limits move according to the season (e.g. the maximum latitude for the detection of ridges is 45º N in winter and 50º N in summer). The optimal latitudinal break relies on previous work about winter climatologies of ridges in the eastern Atlantic (Santos et al., 2009b) and subsequent sensitivity analyses to calibrate their seasonal frequencies in the area of study (Sousa et al., 2016b). Two conditions are imposed for the detection of subtropical ridges: a ridge is detected in a south box if (i) at least 75 % of the grid points are above the 80th percentile of the specific 31-day moving average Z500 climatology during the period 1950-2012 and (ii) no more than 50 % of the grid points of the north box are above the same threshold.

Three longitudinal sectors of 30˚ width with two boxes each (north/south) are used to catalogue high-latitude blocks and subtropical ridges regionally according to the position of their centres (Figure 2): Atlantic (ATL, 30º–0º W), European (EUR, 0º–30º E), and Russian (RUS, 30º–60º E). Note that some blocks and ridges can have their centres in more than one sector during their lifetimes, since they tend to move eastwards. Therefore, the use of specific longitudinal sectors is useful to assess how the location of blocking/ridge centres as well as their residence times impact ozone over different areas of the European continent. The total number of days in which block and ridge centres have been identified over each sector and season are displayed in Table 1 and Table 2, respectively. As more than one block and ridge can be simultaneously identified within different sectors on the same day, the total number of days with blocks and ridges in the region of study does not coincide with the totals shown in the tables. During the 5479 days of the 1998–2012 period, we have computed a total

number of 1609 days with at least one block in any of the three sectors, 1771 days with ridges, and 3050 with blocks and/or ridges. Therefore, more than half the days register at least the occurrence of a block or a ridge anywhere in the whole region. The interannual variability of the number of days with blocks and ridges over a given sector can be rather large, as shown in Supplementary Figure S1.

The seasonal composites of daily Z500 anomalies (with respect to the climatology) and absolute values, considering days with blocking and ridge centres in each sector for a ~60-year period, can be found in Sousa et al. (2016a, 2016b). For illustration purposes, only the composites of daily Z500 for block and ridge centres over the EUR sector during the 1998–2012 period are shown in Figure 3, while Supplementary Figures S2 and S3 display the results for the ATL and RUS sectors, respectively. Stippling indicates statistically significant anomalies at the 5% level (two-sided t-test). The highest values of

the absolute Z500 field (black contour lines) and the positive Z500 anomalies (shaded areas) are centred in the respective sector. Significantly positive anomalies dominate the northern latitudes in the case of blocks and the low latitudes in the case of ridges, being the largest found for blocks as well as in winter for all sectors. Another feature that can be seen on the plots is the northward displacement of the atmospheric circulation from spring to summer, as indicated by the position of the 560 dam isohypse in those seasons (thick back contour lines).

## 2.3 Data analysis

Some of the analyses in the following sections are also based on seasonal composites of MDA8 $O_3$ over Europe. For each season, composite differences are calculated between days with blocks or ridges whose centres are in a specific sector (ATL, EUR or RUS) and days without blocks or ridges in that sector. Moreover, to better quantify the impact of blocks/ridges on

key European regions, probability density functions (PDFs) of MDA8 $O_3$ have been computed for the grid cells within the grey boxes in Figure 2 (from west to east and from north to south: 'UK/North Fr', 'Central/North EU', 'NE Scandinavia', 'Iberia', 'Central/South EU', 'East EU') under different weather regimes. The PDF plots will display ozone data in a given region and season, separately for the whole 1998-2012 climatology and for the days with blocks/ridges in each of the three sectors. The two-sample Kolmogorov-Smirnov test (two sample K-S test; see e.g. Wilks, 2011) has been used to determine

whether the distributions of the daily $O_3$ mixing ratios averaged over such regions on days with blocks/ridges differ from those of the seasonal climatology; we will discuss the situations when the distributions are statistically significant at the 0.1 % level, which may seem too stringent but ensures that only the most relevant changes in the $O_3$ distributions are identified. In addition, to assess the potential implications of blocks and ridges on the occurrence of ozone extremes, days with high ozone extremes are identified locally over each grid cell of the $O_3$ dataset as those when the MDA8 $O_3$ mixings ratios exceed

the 90[th] percentile of the distribution during the corresponding season, while days with low ozone extremes are those below the seasonal 10[th] percentile. Finally, we have tested some multiple linear regression models in order to quantify to what extent the interannual variability of $O_3$ can be driven by the frequency of occurrence of blocks and ridges.

## 3 Impact of high-latitude blocks and subtropical ridges on MDA8 O$_3$ during summer and spring

### 3.1 Regional responses to high-latitude blocks

First, we focus on the impact of blocks on ozone. From the three 30º longitude sectors used in this study, blocking over the EUR sector has the strongest impact on ozone both in spring and summer. On average for the 15-year period of analysis,

MDA8 O$_3$ mixing ratios increase by 5–9 ppb over large areas of central Europe in spring (Figure 4, upper-left) and by 5–11 ppb over northern Europe in summer (Figure 4, upper-right) on days with EUR blocking, which is broadly consistent with the anticyclonic circulation and continental advection observed over those regions. If in the calculation of the composites we only consider data from the third day blocking centres have stayed over the EUR sector, the extension of the regions with anomalies higher than 5 ppb increases and there are larger areas where they exceed 10 ppb (Figure 4, bottom panels). This

minimum number of 3 days is somewhat arbitrary, but it has been chosen to illustrate that the persistence of blocks over a given area favours the photochemical build-up of O$_3$ both in spring and summer. The strongest positive anomalies cover large areas of the continent (40º–60º N) in spring and are located in the north (50º–70º N) during summer, season when weaker negative anomalies are also found in southern Europe. The differences in the latitudinal position of the positive anomalies are due to the northward displacement of the atmospheric circulation from spring to summer. Another factor that

might contribute is the fact that the days are long in summer at high latitudes, which would favour the photochemical build-up of ozone under blocking conditions, but further analyses would be needed to corroborate this hypothesis. The negative ozone anomalies in southern Europe during summer (right panels of Figure 4) are associated with increases in cloud cover and 850 hPa wind speed as well as decreases in shortwave radiation and daily maximum temperature over most of this region on days with EUR blocks (not shown).

The PDFs of MDA8 O$_3$ are illustrated in Figure 5 for both spring (upper panels) and summer (bottom panels). The percentage frequency of all the MDA8 O$_3$ data polled over each region in a given season during the 15-year period is represented by the grey bars, while data on days with ATL, EUR and RUS blocking are represented by the blue, red and green solid lines, respectively. The vertical dotted lines indicate the 10[th] and 90[th] percentiles of the climatology, while the short vertical coloured lines on the top of the plots correspond to the values of the 10[th] and 90[th] percentiles for the data under

the influence of blocking. Figure 5 highlights the increase in the percentage occurrence of high ozone levels over a number of regions in spring and only over the northern regions in summer on days with EUR blocking (red lines), which is consistent with the results from Figure 4. The largest increases in the 90[th] percentile of the ozone volume mixing ratios occur under the influence of EUR blocking for 4 out of 6 boxes in spring and for the 3 northernmost boxes in summer, the differences with the climatology being around 5–10 ppb (compare short solid red lines on top of the panels with the vertical dotted lines).

These changes are mainly due to a shift and to some extent also a broadening of the frequency distribution for EUR blocking days. The results from the two-sample K-S test reveal that the O$_3$ distribution on days with EUR blocks differs from that of the seasonal climatology (at the 0.1% level) for all regions except Iberia in spring and for the three northern regions in

summer. Another emergent feature is the increase in the occurrence of MDA8 $O_3$ mixing ratios within the range 35–55 ppb over 'UK/North Fr' in summer on days with ATL blocking (blue line); however, the two-sample K-S test does not indicate significantly different distributions at the 0.1% level for this particular case because of the small sample size (only 27 days with ATL blocks in summer, see Table 1). Additional analyses have confirmed that this $O_3$ increase is mainly due to the

contribution of days with blocking centres over a narrow longitudinal sector within the East Atlantic (15°–0° W ), when the British Isles are under the influence of an anticyclone (supplementary Figure S4). This synoptic situation, albeit relatively uncommon (only 13 days identified in summer during the whole 15-year period), is efficient at increasing ozone in the British Isles and the northwest of France, because it is associated with a decrease in cloud cover and therefore an increase in the downward shortwave radiation flux, as well as increases in daily maximum temperature and decreases in 850 hPa wind

speed over those regions (not shown); there are also weak negative ozone anomalies in southern Europe, in particular over Italy, where the meteorological anomalies are of opposite sign. Overall, RUS blocking does not seem to be closely related to changes in the PDF of ozone for any region.

The previous analyses have illustrated that significant changes in the mean distribution of MDA8 $O_3$ occur mainly in northern and central Europe on days with EUR blocking during summer and spring, respectively. We have also evaluated the

evolution of the occurrence of above average $O_3$ mixing ratios and extreme ozone days under the influence of EUR blocking. In particular, during summer the extension of the areas with positive ozone anomalies above 5 ppb increases day after day when block centres are in the EUR sector (Figure 6). On average for all blocks remaining in that sector, the fifth day registers the maximum anomaly over any given $1° \times 1°$ cell in the domain (~23 ppb) as well as the maximum extension of anomalies above 5 ppb (~$4 \times 10^6$ km$^2$). Moreover, the maximum extension of the areas exceeding the 90th percentile is found

for that day as indicated by the grey open squares in the figure. Note, however, that only 7 blocks have been considered for the calculation of anomalies on the fifth day compared to 32 on the first day (see numbers in brackets on the top of the panels); the reason for this is that blocks do not necessarily stay over the same sector during their lifetime. A similar evolution has been found for spring, although the overall magnitude of the anomalies and the total area exceeding the 90th percentile are smaller in that season (not shown).

To better illustrate how EUR blocking impacts the occurrence of extremes in summer and spring, we have calculated the percentage of EUR blocking days with MDA8 $O_3$ values above the 90th percentiles of those seasons. The left panels of Figure 7 indicate the likelihood that EUR blocking is collocated with exceedances of the 90th percentile in summer (upper left) and spring (lower left). The results at any location can be compared with what would be expected by random chance (i.e. only 10 % of exceedances of the 90th percentile). More than 30 % of the EUR blocking days concur with exceedances of

the 90th percentile over large parts of Scandinavia, the North Sea as well as the north of the UK in summer. Note however that as ozone levels are low in northern Europe during summer (see middle panel of Figure 1) these events do not often result in exceedances of the EU target value for maximum daily 8-hour mean ozone (120 μg m$^{-3}$, approximately 60 ppb; EEA, 2015). The impact is smaller in spring, when 25 % of the EUR blocking days are coincident with exceedances of the 90th

percentile over large parts of Germany, north-eastern France, the Benelux and the North Sea. Therefore, EUR blocks increase the occurrence of ozone extremes over some regions mainly in summer but also in spring.

The use of 30° longitude sectors, as done throughout most of this work, is useful to generalise the interpretation of the impacts of blocks on ozone, but one could expect the occurrence of extremes to be higher for some areas of the continent if

we only considered blocks with centres over narrower longitudinal sectors. In fact, the areas with the highest occurrence of extremes in the left panels of Figure 7 move westward/eastward when block centres are considered within the west (0° –15° E) / east (15° –30° E) flanks of the EUR sector and the percentage of occurrence of extremes also increases over those areas (not shown).

## 3.2 Regional responses to subtropical ridges

The impact of subtropical ridges on surface MDA8 $O_3$ during spring is considerably smaller than that of blocking for most of the regions considered in this study, while in summer the effect is important for the south and centre of the continent (Supplementary Figure S5). Only the most relevant results for regions where the signal of ridges is strong enough are discussed here, separately for spring and summer.

There are two particular cases that stand out in spring: the Iberian Peninsula under the influence of Atlantic ridges and the

'Central/South EU' region on days with European ridges. These PDFs are respectively shown for the mentioned boxes as blue and red dashed lines in the upper panel of Figure 5. In both cases there is an increase in the frequency of events with MDA8 $O_3$ above ~50 ppb, but the distributions are not significantly different from their corresponding climatological distributions at the 0.1 % level. In the first case, the Azores high moves eastwards and has its centre of action close to Iberia, while in the second case the centre of the anticyclone is located over Italy (not shown). Our findings illustrate that the

regional responses of surface ozone to subtropical ridges are generally more localized than those of blocks and hence they are smoothed if large longitudinal sectors are considered for the analyses. Thus, when considering only ridge centres located over a narrower 15° sector covering the East Atlantic (15°–0° W; see Figure 8, upper left), the centre of the anticyclone is particularly close to Iberia and the isobaric gradient is stronger than the climatology in the north, yielding moderate positive ozone anomalies of up to ~7 ppb over Iberia and small negative anomalies in northern Europe. A similar situation, with

positive anomalies reaching maximum values of ~8 over northern Italy and extending further into the continent, as well as negative anomalies in northern Europe, is seen for ridges in the west flank of the European sector (0°–15° E; Figure 8, upper right). In both cases, the relatively small negative ozone anomalies in northern Europe (only up to around -3 ppb) are associated with increases in 850 hPa wind speed and cloud cover as well as decreases in shortwave radiation over these regions, while the impact of temperature changes does not seem to be relevant.

In summer, the surface $O_3$ increase over the Iberian Peninsula on days with ATL ridges is stronger than that found for the same synoptic situation in spring (Figure S5). Nevertheless, the most prominent feature in this season is the shift towards higher values of MDA8 $O_3$ in the PDF for the 'Central/South EU' box, and to a lesser extent in the 'East EU' box, with the

presence of ridges in the European sector. These three cases are indicated by blue and red dashed lines in the lower panel of Figure 5, all of them being significantly different from the climatology at the 0.1% level. Again, the maximum positive anomalies in those regions are observed when considering ridge centres located over narrow sectors covering the East Atlantic (15°–0° W; Figure 8, lower left) and western Europe (0°–15° E; Figure 8, lower right). The first situation is associated with large ozone anomalies well above 10 ppb in Portugal and negative anomalies in Eastern Europe; overall, the largest positive/negative anomalies seen in the figure seem to be related to increases/decreases in daily maximum temperature over the west/east of the continent, but the impact of other meteorological fields may also play a role. The second situation yields average MDA8 $O_3$ increases above 10 ppb over a larger region, including northern Italy and the surrounding countries, also causing exceedances of the 90th percentile on more than 30 % of the days for many of these locations (Figure 7, upper right). This increase in ozone extremes is not as strong as that seen for northern Europe on days with EUR blocks during the same season (Figure 7, upper left). Moreover, the importance of the persistence of ridges for the build-up of $O_3$ is not as clear as in the case of blocks (not shown). However, the impact on air quality exceedances is noteworthy. This occurs because the 'Central/South EU' region is subject to higher surface $O_3$ levels than other regions. In fact, this is the only region from those analysed here where the summer 90th percentile exceeds 60 ppb (lower panel of Figure 5) and it also includes the Po Basin, where average MDA8 $O_3$ mixing ratios are ~55 ppb in this season. As a consequence, the occurrence of ridges within 0°–15° E pushes the ozone levels high enough so that more than 40 % of the days with this synoptic situation breach the 60 ppb target over a large area including most of the northern half of Italy, the southeast of France, Switzerland and the southwest of Germany (Figure 9). This failure to meet the MDA8 $O_3$ target even occurs on more than 60 % of the days over most of the Po Basin and above 80 % of the days in a smaller region in the proximity of Milan.

**4 Impact of high-latitude blocks and subtropical ridges on MDA8 $O_3$ during winter**

As introduced in Sect. 2 and also documented by previous analyses (Laurila, 1999), during winter predominantly westerly winds from the Atlantic advect background ozone into the European continent, which is a net sink region for ozone. As a consequence, the impact of blocking on surface ozone is expected to be the opposite of that found in summer. Blocking situations divert the western zonal flow and are associated with stagnation, which impedes the arrival of ozone-rich background air masses from the Atlantic and favours the accumulation of pollutants in the boundary layer, reducing surface ozone levels. This can be seen over the northwest of the continent in the composites of winter MDA8 $O_3$ anomalies for blocking centres remaining in the ATL and EUR sectors (Figure 10, upper and middle panels), and over central Europe for blocking centres in the RUS sector (Figure 10, bottom panel). The anticyclone located above the British Isles under ATL blocking situations (Figure 10, upper panel) diverts the typical westerly flow that is present in the climatology (Figure 1, bottom panel), although only relatively small negative anomalies of up to around -5 ppb are found in that region as well as in other countries of western Europe. The impact of EUR blocking over the UK, northern France and the Benelux is stronger, yielding MDA8 $O_3$ negative anomalies of up to around -10 ppb (Figure 10, middle panel). This occurs because under the

influence of ATL blocking north-western Europe is exposed to air masses which re-circulate over the Atlantic, whilst the position of the EUR blocking high better shields the arrival of maritime air masses and it is concomitant with weak winds (on average from 2 to 4 m s$^{-1}$ lower than the climatology at 850 hPa) which transport ozone-depleted continental air masses to those regions. We have also found that with the latter synoptic situation there are large areas of the continent where cloud

cover decreases up to ~25 %, enhancing the loss of longwave radiation at the surface and decreasing daily mean temperatures up to around 7° C. Such conditions are expected to result in low mixing heights and temperature inversions, thus favouring the accumulation of primary pollutants close to the surface and the titration of O$_3$ by NO.

The PDFs of MDA8 O$_3$ in winter (Figure 11) also highlight the increase in the number of low ozone events within the 'UK/North Fr' and 'Central/North EU' boxes under EUR blocking (red solid lines), and to a lesser extent in the 'UK/North

Fr' box when blocking centres are located over the ATL sector (blue solid line). The three distributions differ significantly from the climatology at the 0.1 % level (two-sample K-S test). In particular, the percentage of the ozone occurrences below 20 ppb (10 ppb) in 'UK/North Fr' associated with ATL and EUR blocking is 24 % and 24 % (25 % and 39 %), respectively. Similarly, there is a shift to the left of the MDA8 O$_3$ PDF over the 'Central/North EU' and 'NE Scandinavia' boxes on days with blocking centres over the Russian sector (green lines, Figure 11), this change being significant at the 0.1 % level only

for the latter region. This situation is mainly found with blocking centres over the west flank of that sector (30°–45° E), which divert the westerly flow and result in weak winds that transport continental air masses from the south to those regions (Figure 10, bottom). The figure shows some areas such as the Iberian Peninsula and the British Isles receiving westerly flow from the Atlantic, but the negative ozone anomalies are relatively small in those regions, while the strong negative ozone anomalies of 5–9 ppb found in a large area around Germany and in a smaller region in the north of Scandinavia have

different origins. The first region is subject to weak MSLP gradients and consequently low wind speeds (on average of 2–4 m s$^{-1}$ lower than the climatology at 850 hPa), which should inhibit the ventilation of the boundary layer and favour the loss of ozone by titration at this time of the year. In the second region wind speed increases us much as 3–5 m s$^{-1}$, but is expected to bring ozone-depleted continental air masses from the south (note that at this time of the year average surface O$_3$ levels in northern Scandinavia are higher than in the south, as seen from the lower panel of Figure 1).

The blocking-related decreases in surface O$_3$ over northern Europe concur with an overall reduction of the winter total ozone column (TOC) as reported by Barriopedro et al. (2010a). Their results obey to different mechanisms, as they found that blocking substantially increases the frequency of extremely low TOC values and ozone mini-holes mainly due to the combined effect of horizontal transport of ozone-poor air and vertical motions in the lower stratosphere.

While the overall response to winter blocking is characterized by O$_3$ decreases, it is worthwhile to mention several regions

that actually experience rising levels of MDA8 O$_3$. For example, ATL blocking is associated with a shift to the right in the PDF of MDA8 O$_3$, resulting in statistically significant changes with respect to the climatology at the 0.1 % level, for the two eastern boxes ('NE Scandinavia' and 'East EU', Figure 11). This is particularly the case for the 'East EU' box, as it is often under the influence of westerly flow which is expected to bring ozone-depleted continental air masses (Figure 1, bottom panel) while it is more exposed to background air masses from the ocean under ATL blocking (Figure 10, upper panel).

The PDFs in Supplementary Figure S6 show the responses of different regions to ATL, EUR and RUS ridges. Unlike in the warm seasons, during winter the effects of ridges in northern regions of Europe tend to be the opposite to those of blocks. As an example, Figure 12 (left) illustrates the MDA8 $O_3$ responses to ATL ridges. Iberia is under the influence of the Azores high and the isobars are tightly packed over the UK and downwind regions. The stable weather conditions in Iberia result in moderate negative ozone anomalies, while the strong zonal flow in the north brings background air masses into the continent and ventilates the boundary layer, yielding ozone increases in excess of 5 ppb over some areas of the British Isles and the Benelux. Anomalies of similar magnitude become more widespread, extending from the British Isles to the northern half of Germany, when ridge centres are located in the east of the ATL sector (15º–0º W, Figure 12, right). Similar features, with moderate ozone decreases in the south (not only 'Iberia' but also the 'Central/South EU' box) associated with week MSLP gradients and horizontal wind speeds at lower levels, as well as ozone increases in the north ('UK/North Fr' and 'Central/North EU' boxes), are patent when ridge centres are located over the EUR sector. The shift towards higher MDA8 $O_3$ mixing ratios within the 'UK/North Fr' and to a lesser extent in the 'Central/North EU' boxes under the influence of subtropical ridges with centres over the ATL and EUR sectors can also be seen in Figure 11 (blue and red dashed lines). The effect of ATL ridges can even be detected for the 'Central/South EU' region (also indicated by a blue dashed line). The five distributions are statistically different from those of their respective climatologies at the 0.1 % level. In particular, the PDFs become much narrower and are displaced to the right for the 'UK/North Fr' box. From the visual inspection of the PDFs it is also evident the completely opposite impact of ATL and EUR ridges (blue and red dashed lines, $O_3$ increase) compared to that of ATL and EUR blocks (blue and red solid lines, $O_3$ decrease) over that area.

The different regional responses of near-surface ozone to blocks and ridges found here for winter are not observed in other seasons. This is in line with the results of Sousa et al. (2016b), who found that while near-surface temperatures increase over different parts of Europe with the presence of both blocks and ridges in summer, they decrease/increase under the influence of blocks/ridges over large parts of the continent in winter. This opposite response also manifests in the exceedance of low and high ozone extremes (10th and 90th percentiles of the winter distribution, respectively). While more than 30 % of the winter days with EUR blocking are associated with ozone levels below the 10th percentile over England and northern France (lower right panel of Figure 7), over 20 % of the days with ATL ridges are concurrent with exceedances of the 90th percentile in the proximity of the North Sea (Supplementary Figure S7).

## 5 Role of blocks and ridges in the interannual variability of $O_3$

So far we have shown that summer $O_3$ mixing ratios in northern Europe increase on days with blocking centres remaining over the EUR sector, while in other parts of the continent they are more sensitive to ridges within the east flank of the ATL sector (15˚–0˚ W) and the west of the EUR sector (0˚–15˚ E). On the other hand, winter $O_3$ in north-western Europe decreases on days with blocks in the ATL and EUR sectors, and increases on days with ridges in the ATL sector. There remains the question of whether these synoptic situations can explain the interannual variability of $O_3$.

We have examined this by trying to predict a number of seasonal ozone statistics as a function of the frequency of occurrence of blocks and ridges within each sector and season. After some sensitivity tests, we have found that the strongest responses are found for the number of exceedances of the local 90th percentiles of summer $O_3$ and for the seasonal mean mixing ratios in the case of winter $O_3$. The former (*summer 90th $O_3$*) can be modelled by Eq (1) as a multiple linear fit on the number of blocks in the EUR sector (*nEUR_blocks*), the number of ridges in the east of the ATL sector (*nATE_ridges*) and the number of ridges in the west of the EUR sector (*nEUW_ridges*) during each summer. The latter (*winter mean $O_3$*) has been fitted as a simpler linear combination of the number of blocks and ridges within the east of the ATL sector (*nATE_blocks* and *nATE_ridges*) during each winter, as seen from Eq. (2).

$$summer\ 90th\ O_3 = a_1 + b_1.nEUR\_blocks + c_1 \cdot nATE\_ridges + d_1 \cdot nEUW\_ridges + \varepsilon \qquad (1)$$

$$winter\ mean\ O_3 = a_2 + b_2.nATE\_blocks + c_2 \cdot nATE\_ridges + \varepsilon \qquad (2)$$

where

$a_1, a_2$: intercepts of both models

$b_1, c_1, d_1, b_2, c_2$: coefficients of the total number of blocks/ridges per season within a given sector

$\varepsilon$: random error

The total number of events considered during the 15-year period are large enough – ranging from 84 cases of summer ridges in the west of Europe to 143 winter days with ridges in the east of the Atlantic – so that the interannual variability of these synoptic situations can be used to predict that of surface $O_3$. Fifteen data points representing the seasonal values in each year have been fitted separately for every grid cell in both cases. The significance of the regression models has been tested by carrying out an analysis of variance (F-test; see e.g. von Storch and Zwiers, 1999) at the 5 % level.

The models have skill in reproducing the interannual variability of $O_3$. The explained variance ($R^2$) is above 40 % over large parts of western Europe in summer (top panel of Figure 13), whereas $R^2$ exceeds 60 % in a more compact area spanning from the British Isles to western Germany during winter (bottom panel of Figure 13). The larger extension of the area with $R^2$ values exceeding 40 % in the case of summer is due to the fact that Eq. (1) considers three synoptic situations which tend to impact surface $O_3$ over different regions of the continent. Actually, the linear fit is significant at the 5 % level over three separate regions which correspond to the main centres of action of these patterns (southern Scandinavia, EUR blocks; western Iberia, ridges in the east of the ATL sector; western-central Europe, ridges in the west of the EUR sector). The power of the second model (Eq. 2) to predict the interannual variability of winter mean $O_3$ over a more localised area of north-western Europe is not surprising either. During that season blocks and ridges in the proximity of the ATL sector are respectively responsible for a considerable fraction of the days with the lowest and highest $O_3$ concentrations around the same region, both of them contributing to deviations from the winter mean values; as a consequence $R^2$ values are large and the linear model is significant at the 5 % level over the whole area. Note that these models are relatively simple and that higher values of $R^2$ can be found, in particular in the east of the continent, if the frequencies of occurrence of blocks and ridges within more longitudinal sectors are included as predictors in Eq. 1 and Eq. 2 (not shown). In addition, the reduced data sample (only 15 years) imposes a limit on the statistical significance of the results shown here.

Our results can be put in the context of previous work which has tried to assess the role of the atmospheric circulation in the interannual variability surface of $O_3$. Pausata et al. (2012) found that the correlation coefficients (R) between monthly ozone mixing ratios and the monthly NAOI exceed 0.50 only at some locations in northern and central Europe in winter, while all correlations get smaller in summer even if a modified NAOI is used. Leibensperger et al. (2008) reported that summertime mid-latitude cyclones are good predictors of ozone pollution days in the eastern US, with negative correlations exceeding -0.50 over most areas. Barnes and Fiore (2013) showed that present day summertime variability of surface ozone depends strongly on the jet stream position over the north-eastern US and that this relationship holds under projected climate change scenarios, in which surface ozone variability follows the poleward shift of the jet ($R^2$=0.69). More recently, Shen et al. (2015) have constructed a multiple linear regression model of mean JJA MDA8 $O_3$ in the eastern US on meteorological variables describing the polar jet frequency and the position of the Bermuda High west edge, finding $R^2$ values from 0.53 to 0.71 depending on the region. Our results should be interpreted with caution because of the smaller data sample (15 years of European $O_3$ observations) compared to that of previous studies in the US (around 30 years of observations and longer for modelling studies), but they indicate that the frequency of occurrence of blocks and ridges can explain a considerable fraction of the total variance in the seasonal ozone levels over some regions of western Europe.

## 6 Summary and discussion

We have used a catalogue of high-latitude blocks and subtropical ridges occurring in three distinct 30˚ longitudinal sectors (ATL, EUR, RUS) covering the Euro-Atlantic region, together with a 15-year gridded European surface $O_3$ dataset, to examine the regional responses of MDA8 $O_3$ to the occurrence, position and persistence of blocks and ridges. The maximum ozone anomalies often follow the longitudinal position of the block and ridge centres. As both synoptic patterns disrupt the atmospheric circulation over large areas far from their centres of action, they also exert a moderate influence on the surface ozone levels in distant regions.

We have found a strong impact of blocks on $O_3$ during spring and summer. In particular, EUR blocking is associated with ozone increases over large areas of Europe in both seasons, with anomalies around 5–10 ppb within the 40˚–60˚ N latitudinal band in spring and further north (50˚–70˚ N) in summer. In addition, over 20 % and 30 % of the days with blocks in the EUR sector are concurrent with exceedances of the 90[th] percentile of the ozone distribution in those regions during spring and summer, respectively. Blocks within the ATL sector also yield moderate ozone increases over the UK in summer.

The impact of subtropical ridges on surface ozone during the warm seasons is significant over the centre and south of the continent, in particular during summer. Nonetheless, this effect is more sensitive to their specific location than in the case of blocks. Ridge centres located in the East Atlantic (15˚– 0˚ W) are associated with positive ozone anomalies well above 10 ppb in Portugal during summer. More importantly, there is a large area including northern Italy and the surrounding countries where summer ozone anomalies exceed 10 ppb, often yielding MDA8 $O_3$ mixing ratios above the 60 ppb AQ target, when ridge centres are located in the west flank of the EUR sector (0˚–15˚ E). It is worthwhile to indicate that the 15-

year gridded ozone dataset used here does not include data in southern Italy. Other datasets of shorter duration but improved coverage over that region, such as the 10-year gridded ozone fields produced by Schnell et al. (2014) or time series at selected sites, could be helpful to investigate the effect of ridges in that region. In addition, the joint examination of synoptic and mesoscale weather patterns is needed to better understand the localized impact of circulation on surface ozone in the Mediterranean.

It is well known that anticyclonic conditions, which occur both with high-latitude blocking episodes and under the influence of subtropical ridges, trap pollutants close to the surface in winter, yielding ozone decreases through the enhanced effect of titration and dry deposition. However, this study has revealed that blocks and ridges produce very different responses in surface ozone over the northwest of Europe during this season. The anticyclonic circulation associated with ATL blocking results in moderate ozone decreases over the UK, whereas the response to EUR blocking is stronger (with negative anomalies between -5 ppb and -10 ppb) over the UK, northern France and the Benelux. With the latter synoptic situation, the air masses are re-circulating around the European continent, which acts as a sink for ozone during this season. Conversely, ATL and EUR ridges yield positive winter ozone anomalies in northern Europe, with values exceeding 5 ppb over the British Isles, the Benelux and some parts of Germany. Whilst moderate ozone decreases are found in some southern areas of the continent, these synoptic situations, in particular ATL ridges, are associated with strong MSLP gradients and enhanced zonal flow around 50°- 60° N, favouring the arrival of background air masses from the Atlantic and the ventilation of the boundary layer. Moreover, while 30 % of the winter days with EUR blocks record ozone levels lower than the $10^{th}$ percentile in England and northern France, above 20 % of the days with ATL ridges coincide with exceedances of the $90^{th}$ percentile around the British Isles and the Benelux. Thus, during winter, blocks and ridges lead to opposite effects in both surface ozone mean anomalies and extremes, particularly in northern countries. These findings might not look relevant from the AQ perspective as threshold values for this pollutant are rarely exceeded at most mid-latitude locations in winter. However, these results have potential implications for primary pollutants and particulate matter (PM), whose response is expected to be the opposite to that of ozone. We anticipate that blocking will result in elevated $NO_2$ and PM levels in north-western Europe during winter, and that the changes in the circulation at those latitudes introduced by the influence of subtropical ridges will ventilate the west of the continent. The impact of these synoptic patterns on winter $PM_{10}$ (particulate matter with aerodynamic diameter up to 10 µm) in Europe will be the subject of another publication.

Finally, we have built multiple linear models of some seasonal ozone statistics on the frequency of occurrence of blocks and ridges, proving that these have some skill in controlling the interannual variability of summer and winter ozone over western Europe. The explained variance is considerably larger than that captured by the monthly NAOI for European ozone (e.g. Pausata et al, 2012).

## 7 Implications and future directions

A warming climate is expected to increase peak levels of $O_3$ in polluted regions, an impact that has been referred to as a climate change penalty on AQ. The response of ozone to temperature changes as modelled by both CCMs and CTMs relies on the parameterisation of a number of processes whose temperature dependence is uncertain (e.g. biogenic emissions, which also depend on other environmental factors) or difficult to quantify in practice (e.g. extra evaporation of anthropogenic NMVOCs under high temperatures). Moreover, in regions with a significant contribution of biogenic emissions, ozone-temperature relationships are strongly affected by major uncertainties in the chemistry of isoprene within the models, in particular the isoprene nitrate yields and the fate of these nitrates. Further details on these and other issues can be found e.g. in Vautard et al. (2005), Jacob and Winner (2009), Fiore et al. (2012), Fu et al. (2015) and Zhang and Wang (2016). In addition, it is important to quantify the contribution of both thermodynamic (temperature and specific humidity) and dynamic (circulation) effects to future climate changes (e.g. Shepherd, 2014; Horton et al., 2015). Taking all this into account, future air pollution episodes cannot be modelled as all else being equal except for a uniform temperature shift. As a consequence, the Fifth Assessment Report (AR5) of the Intergovernmental Panel on Climate Change (IPCC) indicates that projecting AQ empirically from a mean surface warming using observed correlations of air pollutants with temperature may be problematic, while establishing relationships with synoptic conditions may be more robust (Kirtman et al., 2013).

Following the IPCC's recommendations, the results from this work provide observational constraints for the process-oriented evaluation of modelled $O_3$ to changes in the occurrence and position of two types of synoptic patterns of great relevance for the weather and climate at the mid-latitudes. We have (i) quantified to what extent these patterns impact the day-to-day variability in the ozone concentrations and (ii) proved that their frequency of occurrence can explain a considerable fraction of the interannual variability of $O_3$ over some areas of western Europe. Therefore, these patterns may serve as dynamical predictors of future ozone variability over this region and can be taken into account when projecting effects of climate change on this pollutant.

It should be borne in mind that a number of studies have shown that climate models often underestimate blocking activity in different regions of the Northern hemisphere (e.g. Scaife et al., 2010; Barriopedro et al., 2010c; Barnes et al. 2012; Anstey et al., 2013; Dunn-Sigouin et al., 2013), with potentially different origins of the model biases depending on the region and season (e.g. Vial and Osborn 2012; Zappa et al., 2014). The ability of some models to simulate blocking is improving, but future trends in blocking are still uncertain (Christensen et al., 2013). Note also that the frequency of blocking may vary across different blocking detection methods and among different reanalysis datasets (e.g. Barnes et al., 2014). To the authors' knowledge, there have been no attempts to evaluate ridge activity over the North Atlantic European region in climate models.

Despite the mentioned uncertainties, our results provide quantitative assessments of the response of surface ozone to circulation changes which can be exploited to evaluate CTMs, as these are driven by meteorological reanalyses like the one used in this work. In this context, the problem with the underestimation of blocking activity in climate models can also be

overcome to some extent by the use of ensembles of historical CCM simulations, which would provide a high enough number of blocking/ridge events whose impacts on surface ozone could be compared with our observational assessments. The analyses conducted here for European ozone can also be extended to other mid-latitude regions and other pollutants like PM in a relatively easy manner. Future efforts in this direction, in combination with improvements in the representation of blocking and ridge activity in climate models, will help understand the contribution of dynamic effects to near-term projections of AQ.

## 8 Data availability

The data can be obtained from the authors of this paper.

## Acknowledgements

Carlos Ordóñez acknowledges funding from the Ramón y Cajal Programme of the Spanish Ministerio de Economía y Competitividad under grant RYC-2014-15036. NCEP Reanalysis data are provided by the NOAA/OAR/ESRL PSD, Boulder, Colorado, USA, from their web site at http://www.esrl.noaa.gov/psd/.

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

**Table 1: Total number of days with blocking centres identified over each sector (Atlantic, ATL, 30º–0º W; European, EUR, 0º–30º E; Russian, RUS, 30º–60º E) and season (winter, DJF: December, January, February; spring, MAM: March, April, May; summer, JJA: June, July, August; autumn, SON: September, October, November) during the 5479 days of the 1998–2012 period.**

|       | ATL | EUR | RUS | Total |
|-------|-----|-----|-----|-------|
| DJF   | 228 | 151 | 75  | 454   |
| MAM   | 200 | 203 | 131 | 534   |
| JJA   | 27  | 91  | 165 | 283   |
| SON   | 103 | 169 | 93  | 365   |
| Total | 558 | 614 | 464 | 1636  |

**Table 2: As Table 1 but for ridges.**

|       | ATL | EUR | RUS | Total |
|-------|-----|-----|-----|-------|
| DJF   | 204 | 199 | 176 | 579   |
| MAM   | 160 | 192 | 156 | 508   |
| JJA   | 184 | 189 | 217 | 590   |
| SON   | 161 | 165 | 142 | 468   |
| Total | 709 | 745 | 691 | 2145  |

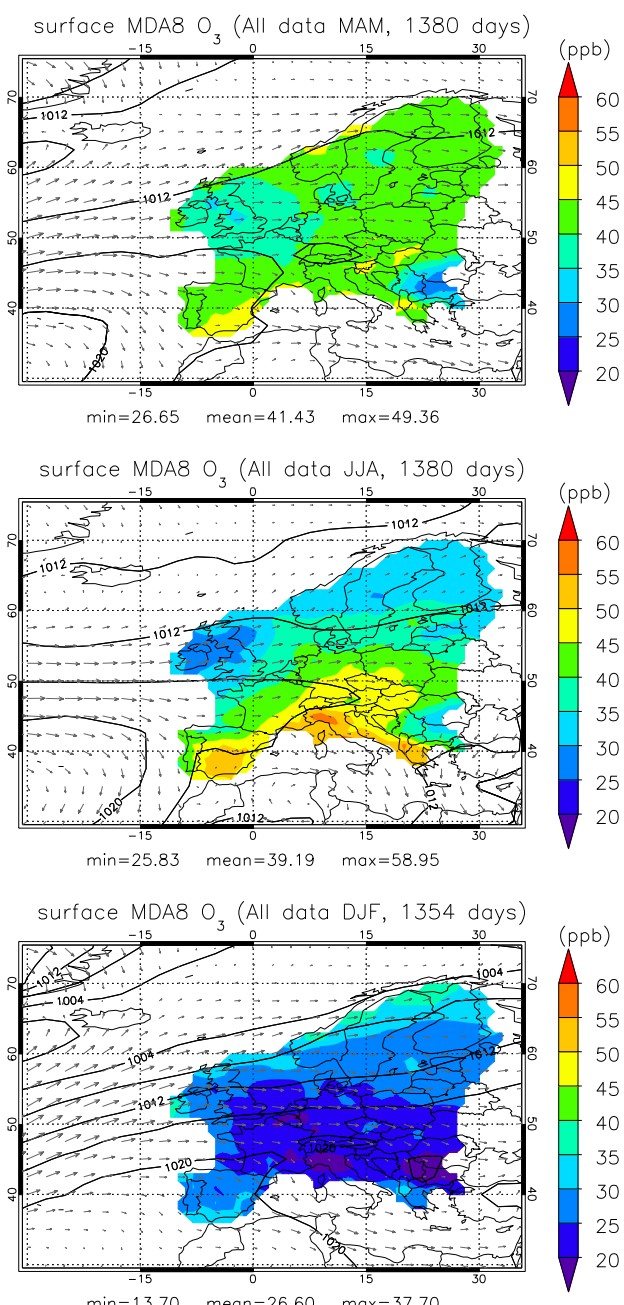

**Figure 1: Composites of the seasonal means of maximum daily 8-hour running average near-surface ozone (MDA8 O₃) expressed in parts per billion by volume (ppb, shaded areas) in spring (top), summer (middle) and winter (bottom) during the 1998–2012 period. The black contour lines depict the mean sea level pressure (MSLP) for the same period, expressed in hPa. Arrows indicate horizontal winds at 850 hPa. The total number of days considered is indicated on the top of each panel.**

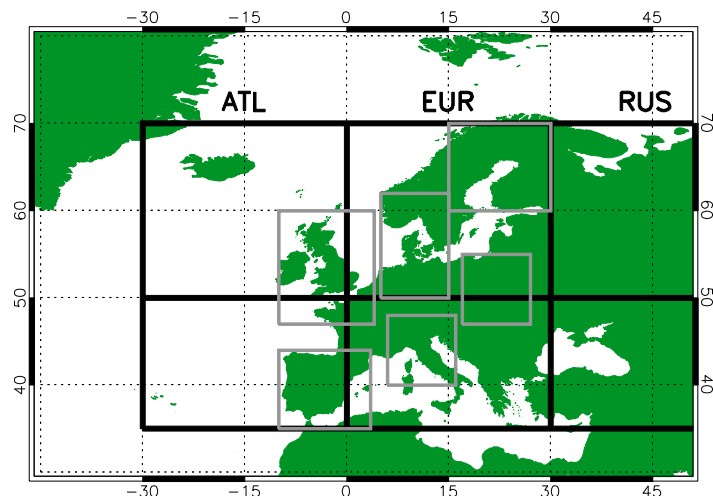

**Figure 2: Longitudinal sectors considered for the identification of the centres of high latitude blocks (northern black boxes) and subtropical ridges (southern black boxes): Atlantic (ATL, 30º–0º W), European (EUR, 0º–30º E) and Russian (RUS, 30º–60º E, only partially shown). Grey boxes identify the geographic areas considered in the regional assessment performed in Sections 3 and 4. The regions considered, from west to east and from north to south, are 'UK/North Fr', 'Central/North EU', 'NE Scandinavia', 'Iberia', 'Central/South EU' and 'East EU'.**

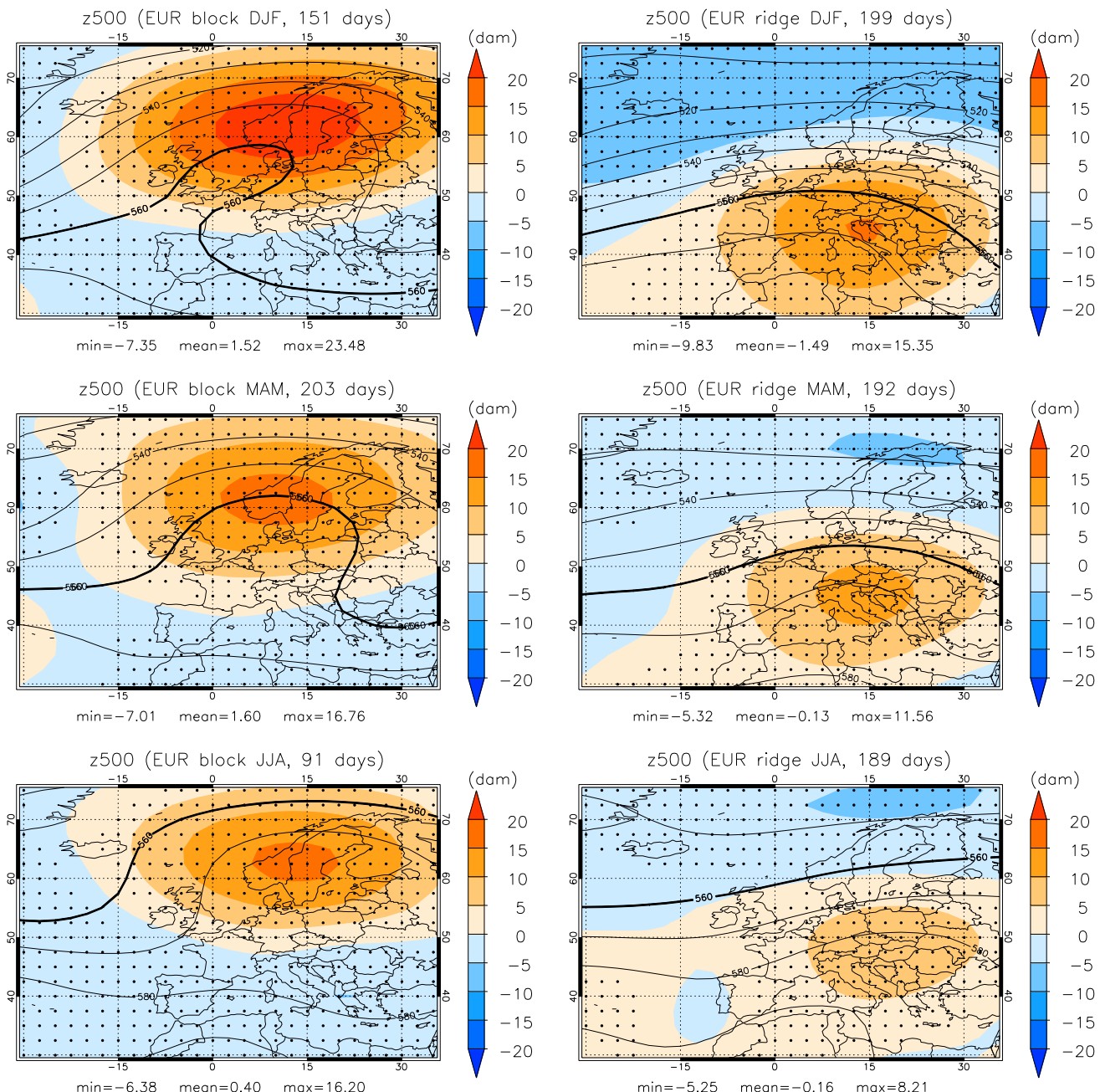

**Figure 3: Composites of the daily anomalies with respect to the 1998–2012 climatology (shaded areas) as well as absolute values (black contour lines) of 500 hPa geopotential height (Z500) for blocking (left) and ridge (right) centres within the European sector in winter (top), spring (middle) and summer (bottom). Stippling indicates statistically significant anomalies at the 5% level (two-sided t-test). All values are in decametres (dam) and the thick line represents the 560 dam isohypse. The total number of days considered is indicated on the top of each panel.**

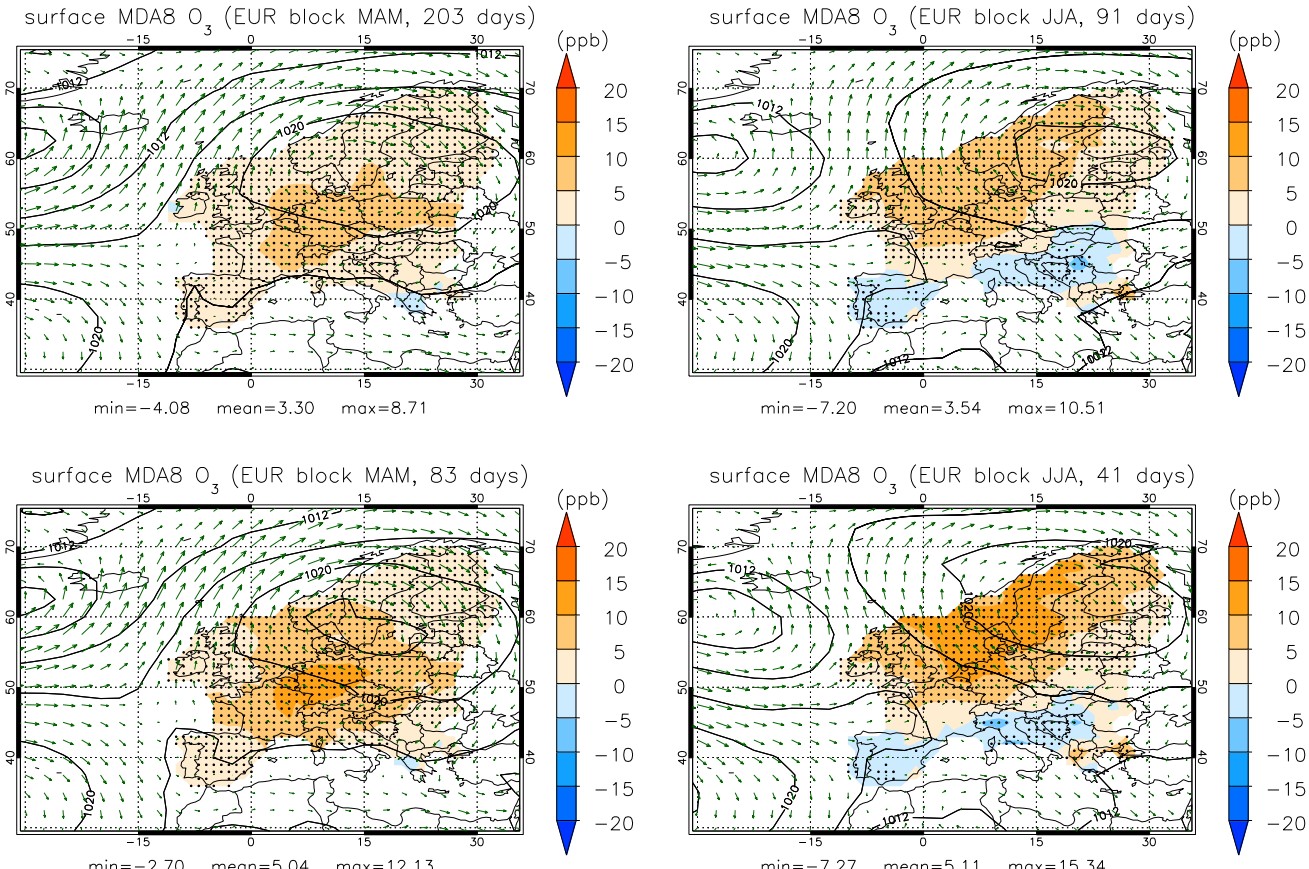

**Figure 4: Composites of the seasonal anomalies of surface MDA8 O₃ (ppb) for days with blocking centres over the European sector (EUR, 0°–30° E), during spring (left) and summer (right). Anomalies have been calculated with respect to the MDA8 O₃ mixing ratios on days without EUR blocking in that season during the 1998–2012 period. All data on days with blocking centres remaining over 0°–30° E (top) and data only from the third day with blocks in that sector (bottom) have been taken into account for the calculation of the ozone anomalies. Stippling indicates statistically significant anomalies at the 5 % level (two-sided t-test). The black contour lines depict the composites of MSLP (hPa) for those days. Horizontal wind fields at 850 hPa are displayed by arrows. The total number of days considered is indicated on the top of each panel.**

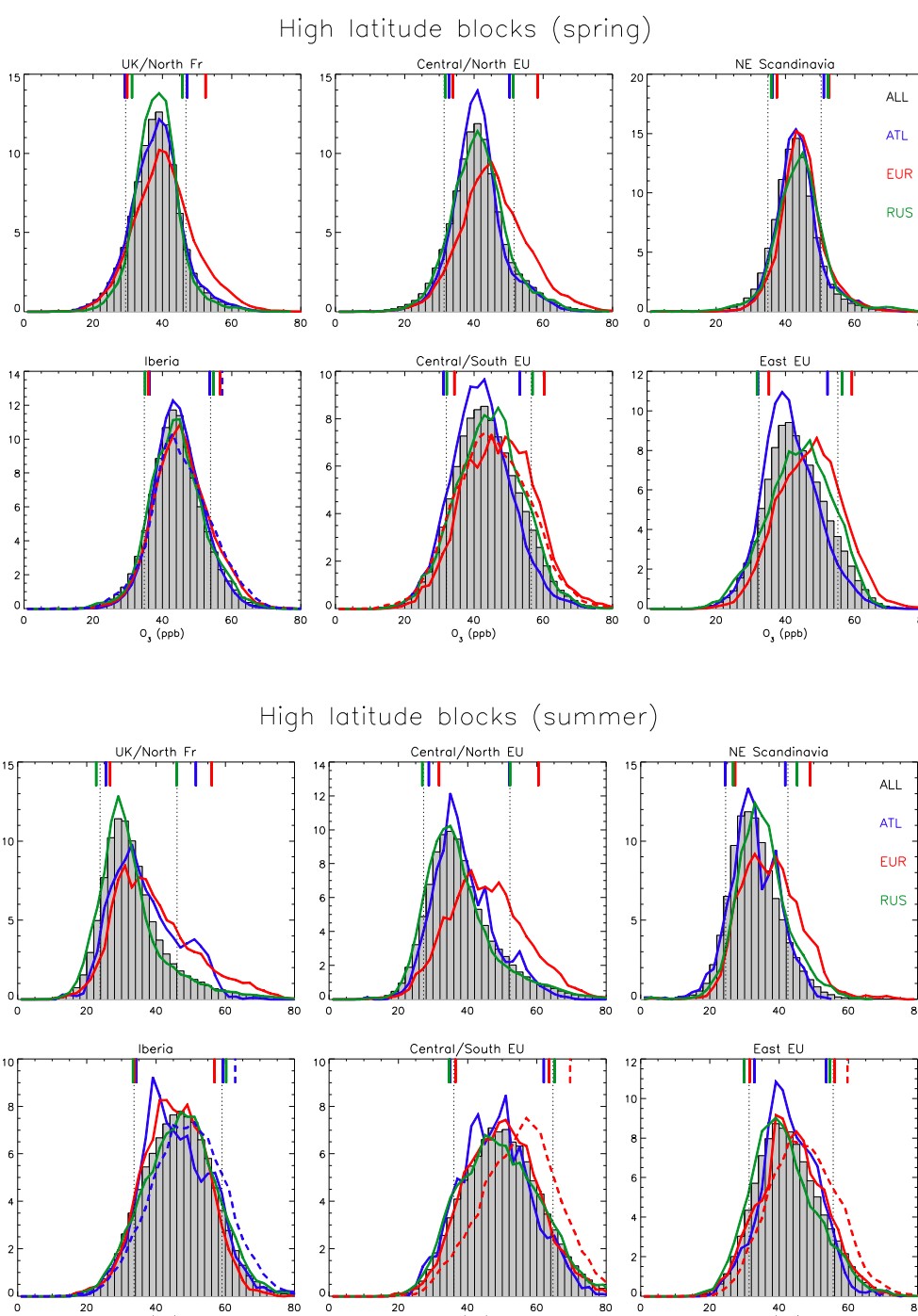

**Figure 5: Spring (upper panels) and summer (lower panels) probability frequency distributions (%) of MDA8 O₃ for the six regional boxes presented in Figure 2 under different synoptic situations. Grey bars denote the 1998-2012 seasonal climatology. Solid lines correspond to days with block centres over the Atlantic (blue), European (red) and**

Russian (green) sectors. Blue and red dashed lines also represent the frequency distributions of the data when ridge centres are located over the ATL and EUR sectors, respectively; they are only shown for the regions and seasons where surface ozone presents the strongest responses to ridges. The width of the bins is 2 ppb. The horizontal position of the vertical dotted lines represents the 10th and 90th percentiles of the climatology, while the short vertical lines on the top of the plots indicate the values of the 10th and 90th percentiles under the influence of ATL, EUR and RUS blocking. The 90th percentiles of the data with ridges are also displayed as dashed lines when they are above those with blocking.

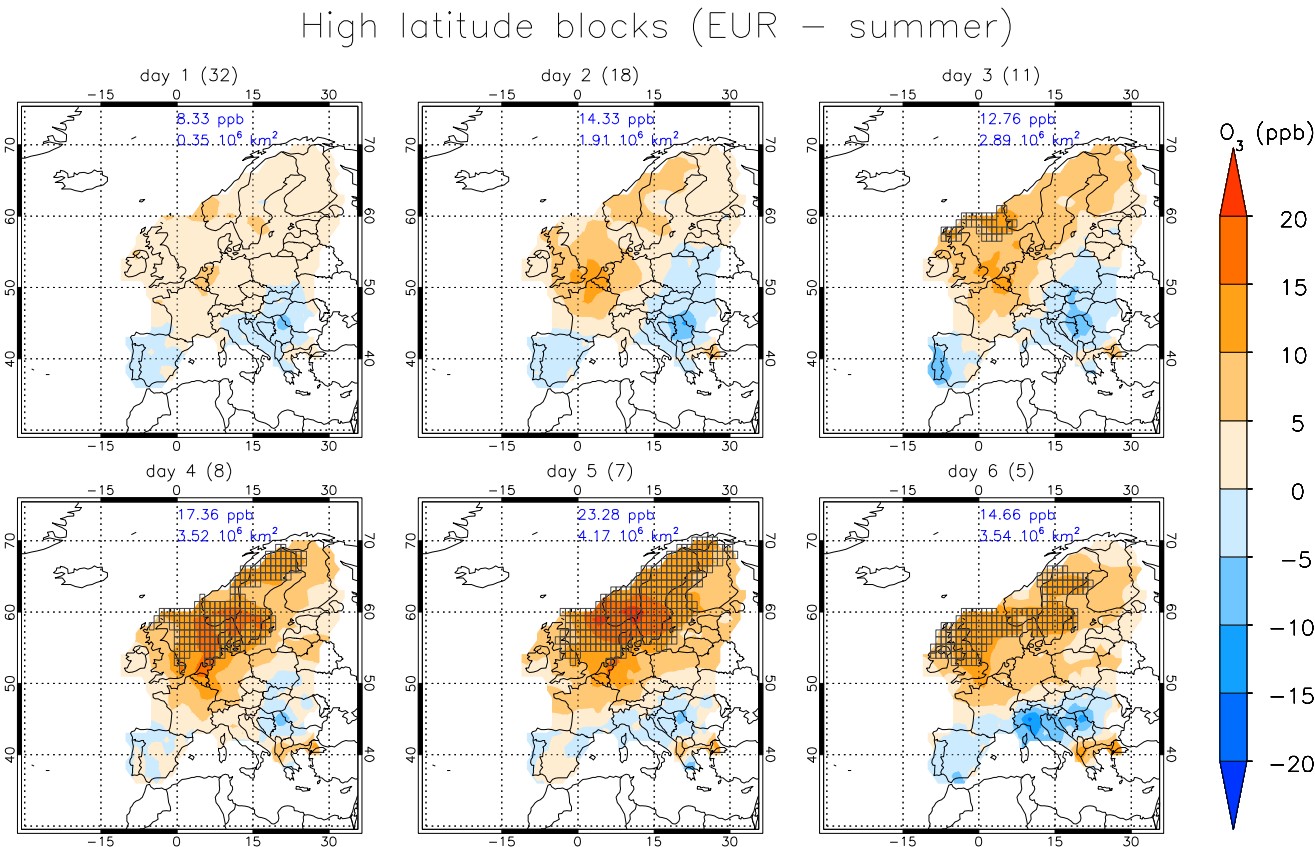

Figure 6: Evolution of MDA8 O₃ anomalies under the influence of EUR blocking in summer. The panels show, from left to right and from top to bottom, the average anomalies on the first six days a block is in the sector. The total number of blocks considered is given in brackets on the top of each panel. The maximum anomaly found (ppb) and the total area with anomalies higher than 5 ppb (km²) are also indicated in blue colour. Grey open squares indicate grid cells where the ozone composite under blocking exceeds the 90th percentile of the summer distribution.

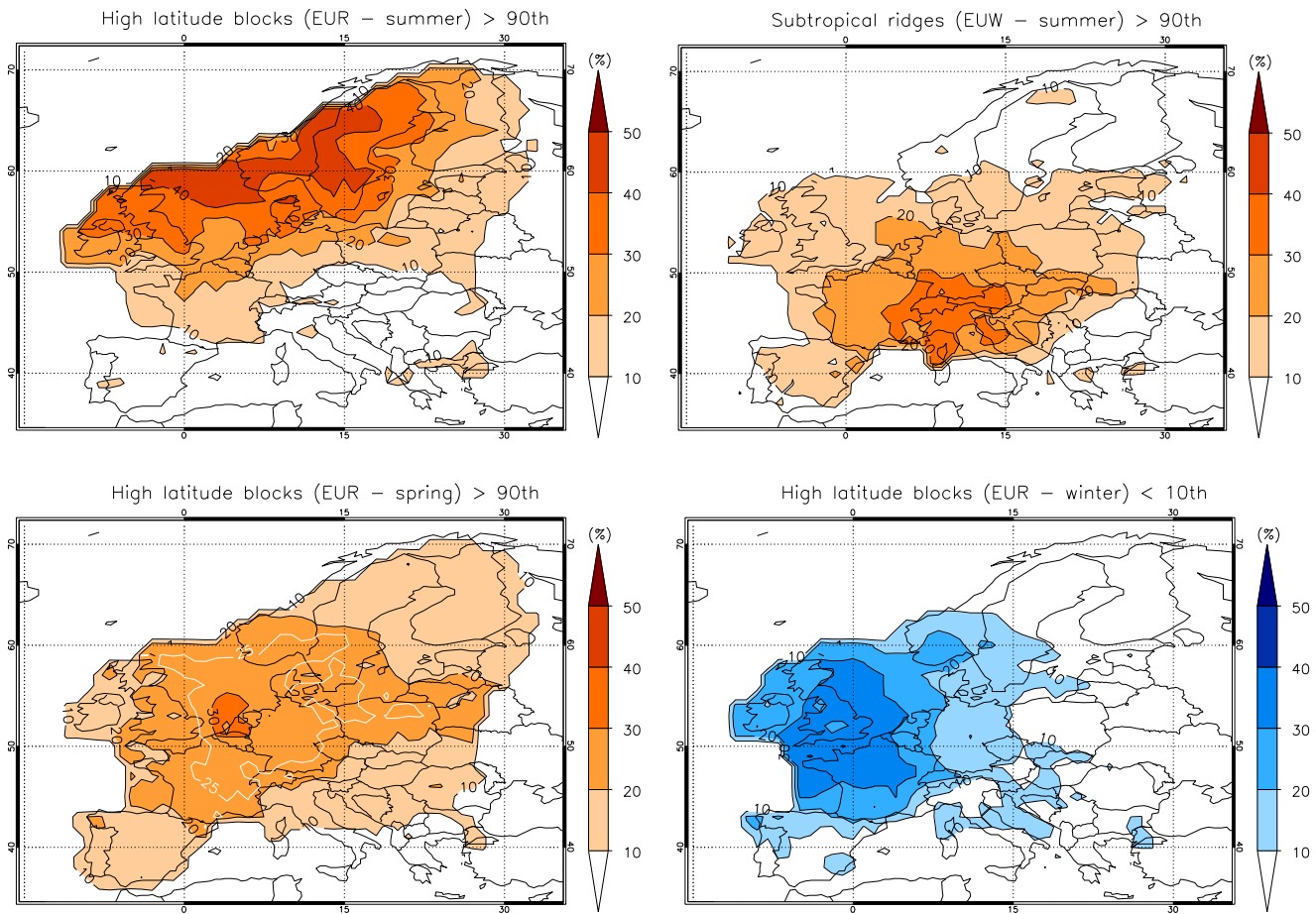

**Figure 7: Percentage of EUR blocking days in summer (upper left) and spring (bottom left) as well as of days with ridges over the west of the EUR sector in summer (0°–15° E, upper right) for which MDA8 O₃ is above the local 90th percentile of its seasonal distribution. The lower right panel also shows the percentage of EUR blocking days with MDA8 O₃ below the 10th percentile of its seasonal distribution in winter.**

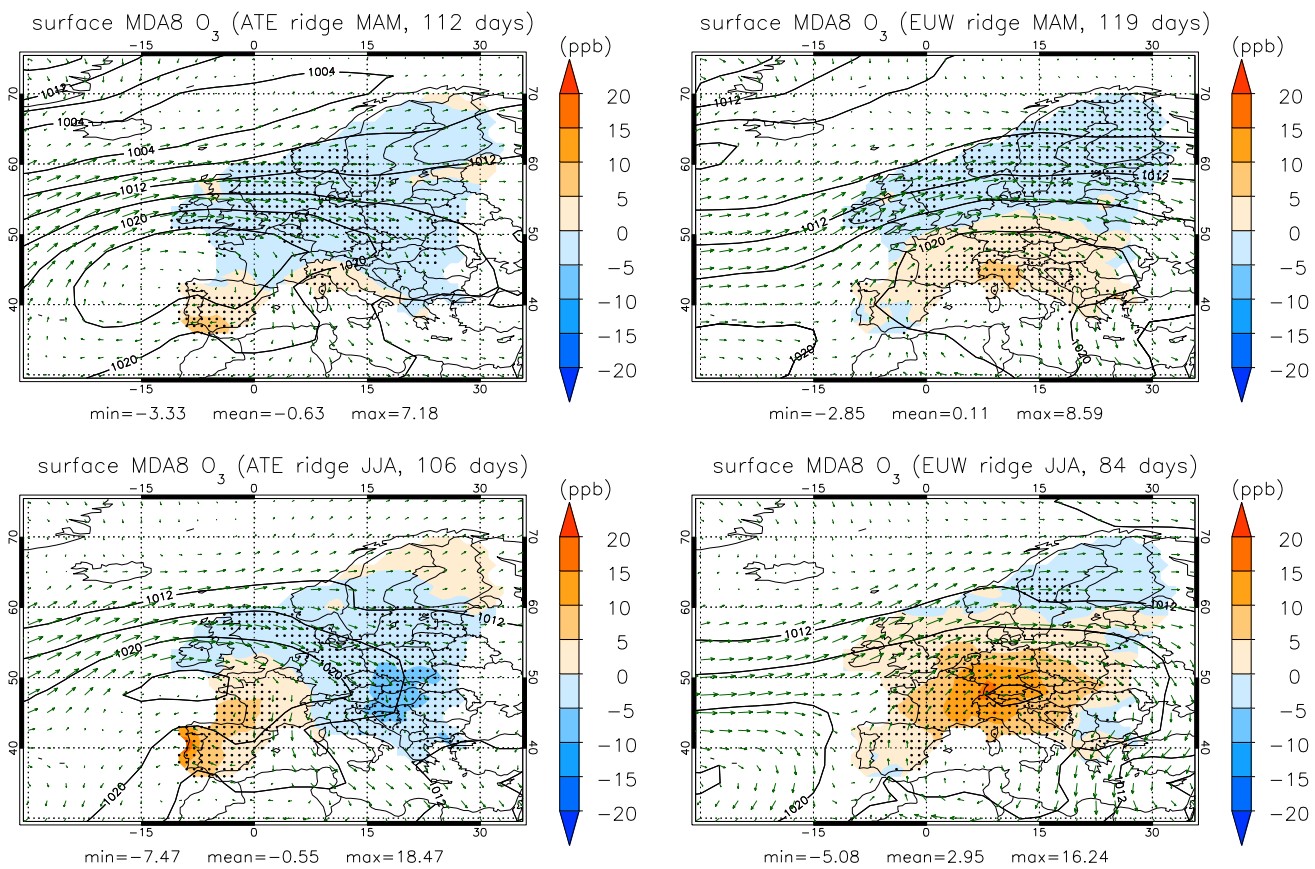

Figure 8: As Figure 4 but for subtropical ridges with centres over the East Atlantic (15°–0° W, left) and the west of the EUR sector (0°–15° E, right) in spring (top) and summer (bottom).

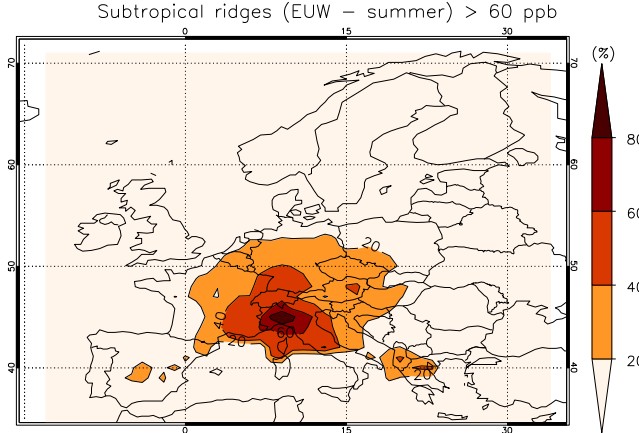

**Figure 9: Percentage of summer days with ridge centres within the west flank of the EUR sector (0-15º E) for which the 60 ppb MDA8 O₃ target is breached. Note the similar geographic location of the region with exceedances on more than 40% of the days in this figure and that where the local 90ᵗʰ percentile is surpassed on more than 30 % of the days under this synoptic situation (Figure 7, upper right).**

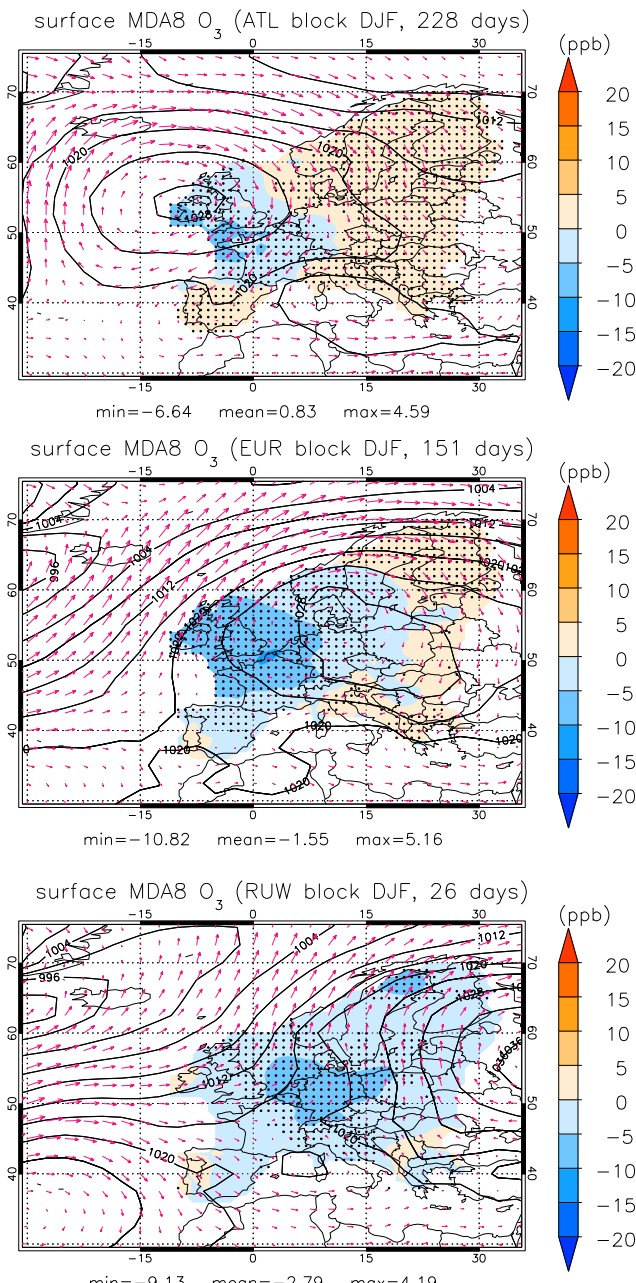

surface MDA8 O$_3$ (ATL block DJF, 228 days)

(ppb)

min=−6.64    mean=0.83    max=4.59

surface MDA8 O$_3$ (EUR block DJF, 151 days)

(ppb)

min=−10.82    mean=−1.55    max=5.16

surface MDA8 O$_3$ (RUW block DJF, 26 days)

(ppb)

min=−9.13    mean=−2.79    max=4.19

**Figure 10: As Figure 4 but for blocking centres remaining in the Atlantic (30°–0° W, top) and European (0°–30° E, middle) sectors as well as in the west flank of the Russian sector (30°–45° E, bottom) in winter.**

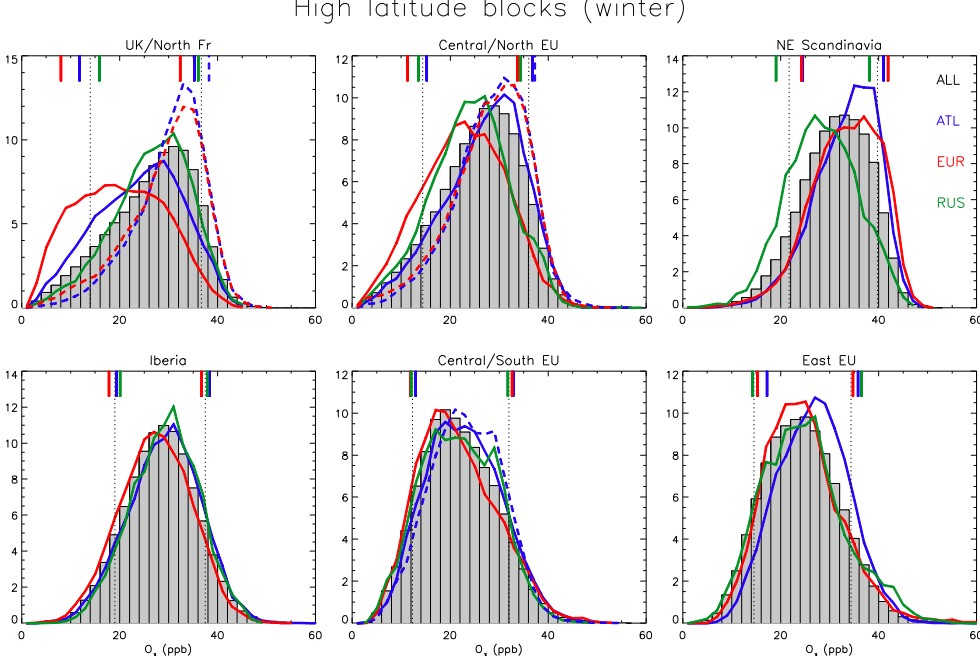

**Figure 11: As Figure 5 but for winter blocking. Note that blue and red dashed lines have been added to represent the frequency distributions of the data within the 'UK/North Fr' and 'Central/North EU' boxes when ridge centres are located over the ATL and EUR sectors, respectively. Data on days with ATL ridges are also shown for the 'Central/South EU' region.**

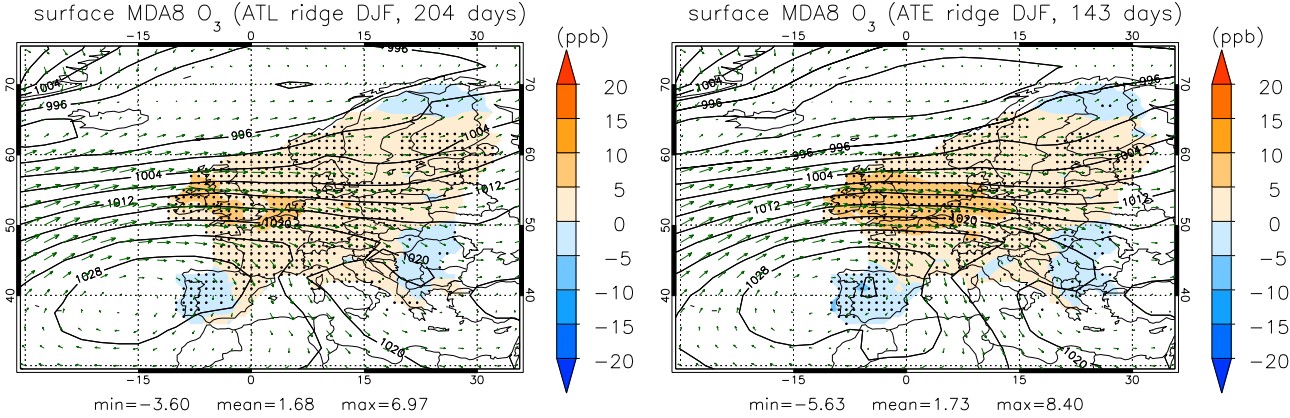

**Figure 12: As Figure 4 but for subtropical ridge centres in the Atlantic (30°–0° W, left) and East Atlantic (15°–0° W, right) in winter.**

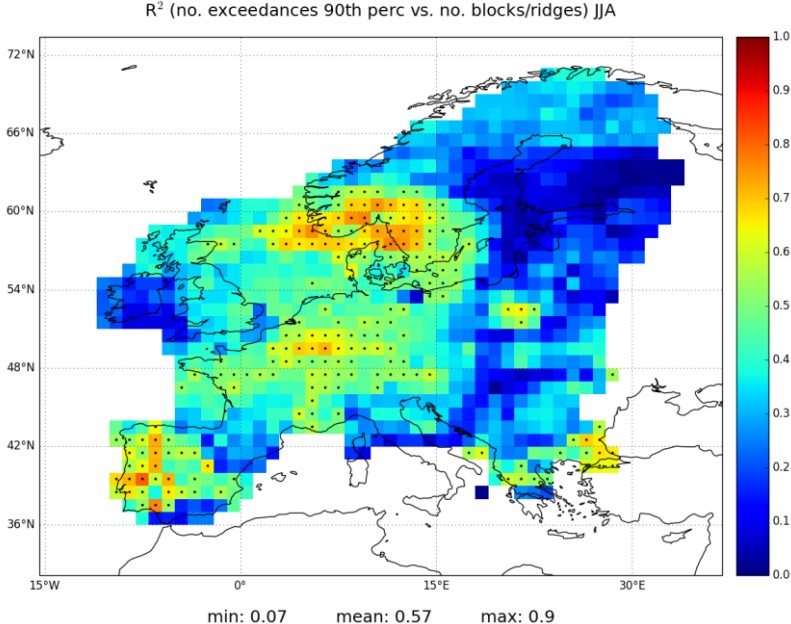

min: 0.07     mean: 0.57     max: 0.9

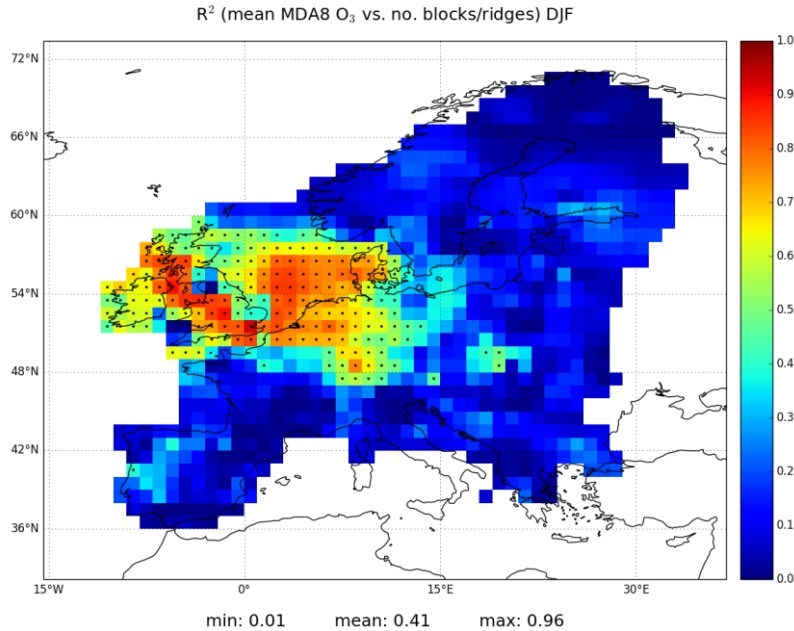

min: 0.01     mean: 0.41     max: 0.96

**Figure 13: Coefficient of determination ($R^2$) of two multiple linear models predicting the number of exceedances of the 90th $O_3$ percentile in summer (top panel, Eq. 1) and the winter mean $O_3$ mixing ratios (bottom panel, Eq. 2) over each grid cell as a function of the interannual variability of blocks and ridges. Stippling indicates the locations where the multiple linear fits are significant at the 5 % level (F-test).**