# Peer review of "Regional responses of surface ozone in Europe to the location of high-latitude blocks and subtropical ridges"

_Atmospheric Chemistry and Physics, 2016_

## Referee Comment (RC1) · Anonymous Referee #1 · 26 Nov 2016

Ordonez et al. present a comprehensive analysis of the influence of the location of high-latitude blocks and subtropical ridges on European near surface ozone concentrations. The catalogue for blocks/ridges based on geopotential height at 500 hPa from the NCEP/NCAR reanalysis and a novel gridded dataset for maximum daily 8-hr average ozone utilized by the authors have been well-documented in the recent literature. The manuscript is clearly structured, well prepared and provides interesting insights in European surface ozone variability. I recommend publication in ACP after some minor clarifications/changes described below.

Specific comments:

1) The authors are using the NCEP/NCAR reanalysis at 2.5° x 2.5° horizontal resolu-

tion to detect high-latitude blocks and subtropical ridges. In Sect. 2.2 we learn that the same catalogue as in Sousa et al. (2016a,b) is used. I wonder if the frequency of blocks and/or ridges identified would significantly differ if another reanalysis (ERA-Interim or MERRA) at higher horizontal resolution would be used?

2) Do the grey bars in Figure 5 really show the seasonal climatology (all data) or do they show surface MDA8-O3 during times when no block/ridge is present in the domain? If all data is shown I suggest adding a PDF curve also for the no block/ridge case.

3) Sections 3 and 4 provide a very comprehensive overview about the effects of blocks and ridges on seasonal MDA8-O3. To allow easier comparison among individual cases I suggest combining Figures 7 and 13 in one 4-panel illustration.

———————————————————

---

## Referee Comment (RC2) · Anonymous Referee #2 · 26 Nov 2016

Ordóñez et al. present an analysis of the impacts of high-latitude blocks and subtropical ridges on surface air quality, particularly ozone, in Europe. The authors use a previously published catalog of these meteorological features to study spatial and temporal changes in ozone concentrations. The connections are made clear and the statistical analysis is sound. The manuscript is clear and pretty concise; in general it is very well-written. I believe the manuscript is high quality and requires limited revision. I recommend publication of this manuscript in ACP after the authors' address the following minor comments.

General Comments

- The most significant issue with this manuscript is the lack of a mechanistic discussion

and/or a clear explanation detailing the significance of the results. I disagree that the results are directly usable by modeling communities in assessing changes and validating CCMs. Much more quantitative metrics are necessary.

An attempt is made to broaden the results in the final section, but it is not overly convincing. Indeed, many of the relationships presented are not terribly surprising; the relationship between ridging/blocks and air pollution has been known for decades and even presented within this manuscript as a given. What is the significance of these results? Can a predictive model be developed? Can the results be used to better understand changes brought about by climate change? Can the seasonal frequency or intensity of the EUR blocks be used to summarize seasonal ozone statistics? How can all of this be related to meeting the ozone air quality target? These are the metrics that would make this work more useful to policymakers and model developers alike.

- Regarding the above comment, the authors provide a nice summary of prior meteorology-ozone research in the introduction. It would be very instructive to revisit those studies later in the manuscript since many of the previous connections can be recast or mechanistically explained using these prior studies.

- Why is the spatial aggregation of ozone data (the Schnell data) of better use than observations from individual sites? It seems that the averaging of the observations will mask out some variability and localized features, which may provide valuable mechanistic insights. This is also the case when discussing changes in the PDFs.

- The terminology used in the manuscript is generally spot-on, but I question the use of the terms "ridge" and "blocks". Blocks usually imply a long lasting atmospheric pattern (indeed, a ridge) that actually "block" and divert the normal flow of extratropical cyclones. The term block, in my experience, has a lot less to do with latitude (as used in this manuscript), but rather the persistence. The terminology of ridge vs. block does become cumbersome as the reader progresses.

Can one block or ridge migrate from one "box" to another? If so, how does that influ-

ence ozone?

- A number of negative anomalies, generally in the Iberian Peninsula and Italy. These areas have large values, but go unexplained. I recommend adding some discussion about these areas.

Specific Comments

Page 2, Line 11: The degree of significance used in the confidence intervals is unclear and should be specified.

Page 2, Lines 12-14: The identity of these indirect radiative effects are a mystery to the casual reader. Maybe add a few more descriptive words.

Page 2, Line 18: The more recent review by Fiore et al. [2015] may be a better inclusion.

Page 3, Line 3-5: A daily NAO index is generated and might prove useful. I cannot speak for the results in Sousa et al. (2016), but I would not discount the NAO index simply because of a perceived monthly timescale.

Page 5, Lines 1-4: How trustworthy is the NCEP/NCAR reanalysis I compared to more recent reanalysis projects? Why not use ECMWF-Interim? Or MERRA? Or JRA? All of these have higher spatial resolution and may provide a better picture of the meteorological landscape. Also, the placement of the description of met. data here is curious; it could be moved to the beginning of Section 2.2.

Page 5, Lines 9-11: This is curious. Why is this the case? It is interesting that effort is placed into the winter months since ozone levels are so comparatively low in that season, surely lower than in the autumn.

Page 5, Line 29-31: Low values of ozone do not necessarily indicate that there is a preferred mechanism (decreased formation, increased loss, or both of ozone). This sentence hints at attribution that is not described.

[Figure]

Page 6, Line 7: With all meteorological-air quality studies it is important to know the sensitivity of the meteorological metric to the assumed parameters. How important is this area requirement? What about a weaker, but persistent ridge, which may be missed by this algorithm? How is the "best" latitudinal break between ridges and blocks determined?

Page 6, Line 7: How long does the average block persist? This is important information for the later analysis of the buildup of ozone during an episode.

Page 6, Line 12: What happens if criteria ii is not met? Is it considered a block?

Page 6, Lines 23-25: What is the variability in the frequency of blocks and ridges?

Page 7, Line 23: Isn't the requirement in Section 2.2 that the block must last at least 5 days? If the system is moving so rapidly through the region (< 3days), is it really a "block"?

Page 8, Line 15: The Scandinavian PDF with EUR blocks looks very similar to the climatology, perhaps even more so than Iberia. Is it really significantly different?

Page 8, Line 20: Why is this position of the block so efficient at increasing ozone in UK/N. France?

Page 10, 1st paragraph of section 4.: I think that this needs a bit more thought and discussion. The reason for anomalously low ozone values that is provided suggests the cut off of marine air that brings in background ozone. Yet, Figure 10 shows that locations susceptible to marine air (as indicated by arrows) have negative anomalies, particularly in the RUW case (bottom of Fig. 10).

Page 11, Line 11-13: As mentioned above, this is not what I am seeing. I agree that continental air is advected northward, but the largest anomalies appear to be receiving westerly wind from the Atlantic.

The contour lines within the figures containing maps are very hard to distinguish. I

recommend thickening them up before publication.

Reference

Fiore A. et al., Air quality and climate connections, J. Air Waste Management, 65 (6), 2015.

---

## Author Comment (AC1) · 24 Jan 2017

Ordóñez et al.: Regional responses of surface ozone in Europe to the location of high-latitude blocks and subtropical ridges

(1) Comments from Referees and (2) author's response.

We are thankful to the referees for the time they have taken to review the manuscript and for their constructive comments. Our replies are given after their comments.

We have updated the manuscript following most of their recommendations. In addition, we have introduced the following changes:

* The subtropical ridge detection algorithm has been improved. The ACPD manuscript indicated that "a ridge is detected in a south box if (i) at least 75 % of the grid points are above the 80th percentile of the Z500 climatology of the specific month during the period 1950-2012 and (ii) ...". In the revised version the catalogue of ridges is based on exceedances of the "80th percentile of the specific 31-day moving average Z500 climatology". The use of running means is more appropriate than a monthly mean, in particular during periods with significant changes in the atmospheric circulation at mid-latitudes. Consequently, Table 2 (total number of days with ridges for each sector and season) as well as all figures containing results for ridges have been updated. The use of 31-day moving averages in this improved catalogue reduces the total number of ridges detected for most sectors and seasons, so we have reassessed the impact on the surface ozone concentrations. Overall, the main messages of the manuscript have not changed drastically. However, the ozone responses to ridges are now much more marked in summer and smaller than previously found during winter. Some parts of the text have been updated accordingly. In particular, the text in Section 3.2 (regional responses to subtropical ridges during summer and spring) has changed considerably. Following this, we have also introduced additional changes in some figures:

a) The former Figure 13 on percentage of exceedances of ozone extremes (10th and 90th percentiles) in winter has been moved to Figure 7, as requested by referee #1. However, the panel on ozone extremes for ATL ridges in winter has not been kept there and has been moved to the supplement (Figure S7). Within Figure 7, this has been replaced by an equivalent panel for EUR ridges during summer as this is definitely more relevant now.

b) Figure 8 on the impact of ridges in spring and summer has changed from 2 to 4 panels. This way it is easy to compare how the same synoptic situations impact surface ozone in both seasons (similar patterns but stronger in summer than in spring).

c) We have removed the time series showing the effect of synoptic persistence at specific locations in previous Figure 9 (left panel for Milan in summer and right panel for

Amsterdam in winter), because the effect of persistence does not seem to be so relevant for ridges any more. We have replaced this by a single panel figure showing the strong impact of ridges on air quality exceedances over a region around northern Italy in summer, as this result is clearly more relevant. Note also that the impact of synoptic persistence during summer is discussed for blocks in other parts of the text (e.g. Figures 4 and 6).

d) Figure 12 does not show the impact of the persistence of ridges in winter any more for the reasons explained above. Now the two panels illustrate the sensitivity of ridges to their specific location (within a large 30° longitudinal sector vs. a narrower 15° longitudinal sector).

* The blocking detection method remains unchanged. However, in the revised version of the manuscript we clearly state that it is based on identifying large-scale reversals of the Z500 meridional gradient (not on "large-scale Z500 departures above a seasonally varying threshold" as indicated before).

All changes to the main text and the supplement are highlighted in blue colour.

Anonymous Referee #1

Ordonez et al. present a comprehensive analysis of the influence of the location of high-latitude blocks and subtropical ridges on European near surface ozone concentrations. The catalogue for blocks/ridges based on geopotential height at 500 hPa from the NCEP/NCAR reanalysis and a novel gridded dataset for maximum daily 8-hr average ozone utilized by the authors have been well-documented in the recent literature. The manuscript is clearly structured, well prepared and provides interesting insights in European surface ozone variability. I recommend publication in ACP after some minor clarifications/changes described below. Specific comments:

1) The authors are using the NCEP/NCAR reanalysis at 2.5 x 2.5 horizontal resolution to detect high-latitude blocks and subtropical ridges. In Sect. 2.2 we learn that the

same catalogue as in Sousa et al. (2016a,b) is used. I wonder if the frequency of blocks and/or ridges identified would significantly differ if another reanalysis (ERA-Interim or MERRA) at higher horizontal resolution would be used?

From our experience, there are no substantial differences between the climatological spatial distributions of blocking frequencies when different reanalyses are used with the same (2.5°) horizontal resolution. At least this is true for bi-dimensional blocking detection methods, including the one used here.

We have compared our catalogues of blocks/ridges using both NCEP/NCAR and ERA-Interim data at 2.5° x 2.5° horizontal resolution, and have found that the climatologies are very similar. The differences in the frequencies of blocks/ridges, the Z500 composites, and the corresponding meteorological anomalies are minor.

Note also that previous studies have evaluated the frequency of blocks for different reanalysis datasets. However, their results may look somewhat contradictory and they do not provide much information on the impact of horizontal resolution either. A couple of examples are given here:

* Davini et al. (2012) analysed winter blocking through the introduction of a set of bi-dimensional diagnostics and compared the frequency of winter blocking in the NCEP/NCAR, ERA-40 and ERA-Interim reanalysis data at the same resolution as us (2.5° x 2.5°). They found good agreement between the three datasets, in particular over Europe, even on a daily basis (see summary plots in their Figures 1–3). They only showed results from NCEP/NCAR in all the analyses throughout the rest of their paper because the results were very similar for the other two reanalysis datasets.

* Barnes et al. (2014) evaluated blocking frequencies for three blocking detection methods (one-dimensional and two-dimensional), each of them applied to four different reanalysis datasets (ERA-Interim, MERRA, NCEP/NCAR and NCEP2). No information on the horizontal resolution of the reanalysis data is given in that paper. Unlike Davini et al. (2012), they found different frequencies among the three datasets for a given

blocking index. However, it is difficult to understand the origin of those differences as their results were presented over large longitudinal sectors of 60°. See e.g. right panels of their Figure 3 for North Atlantic blocking (defined as 300ĚŽ-360ĚŽ E) in summer, and their Supplementary Figure 3 for Europe blocking (0°-60° E) both in winter and summer.

To summarise, our blocking and ridge climatologies agree for the two reanalysis datasets mentioned here when they are used at the same resolution, in line with the results of Davini et al. (2012). We have not assessed the impact of horizontal resolution, but blocks are large-scale phenomena well above the synoptic scale (although for simplicity the term "synoptic" is applied throughout the manuscript). Hence, there is no need of using high spatial resolution (as it happens for synoptic systems such as cyclones). In fact, some methods apply a low-pass filter to remove synoptic variability before detecting blocking. Moreover, potential differences may arise by methodological artefacts. For example, there could be differences between ERA-Interim at 1.25° and 2.5° related to the fact that it is more difficult to satisfy the spatial extension condition (reversals in "contiguous" grid points extending over 12.5°) for the 1.25° than for the 2.5° version.

An exhaustive discussion of the impact that different reanalysis datasets (arising from different horizontal resolution, model parameterizations or data assimilation procedures) or even different blocking detection algorithms may have on the frequency of blocking and, consequently, on surface ozone, is out of the scope of this paper. Therefore, all this information will not be added to the manuscript. We have simply added a couple of comments. At the beginning of Section 2.2 ("Climatology of high-latitude blocks and subtropical ridges") we have justified that the horizontal resolution used is appropriate because blocks and ridges are large-scale phenomena. We have also added a sentence to the last section of the paper to acknowledge that results may be sensitive to some choices: "Note also that the frequency of blocking may vary across different blocking detection methods and among different reanalysis datasets

(e.g. Barnes et al., 2014)".

2) Do the grey bars in Figure 5 really show the seasonal climatology (all data) or do they show surface MDA8-O3 during times when no block/ridge is present in the domain? If all data is shown I suggest adding a PDF curve also for the no block/ridge case.

While the composite maps shown in this manuscript display anomalies which are calculated by subtracting data on days with blocks/ridges in specific sectors and data without blocks/ridges in the same sectors, we have preferred comparing the block/ridge data to the whole climatology (all data) for the PDF figures. This was mentioned in the last paragraph of section 2.2 (now Section 2.3 Data analysis). Moreover, the caption of Figure 5 and the main text clearly state that the grey bars correspond to the seasonal climatology (i.e. all data).

We initially considered the possibility of adding PDF curves for the no block/ridge cases. However we decided not to do so because this would not aid the interpretation of the results. These figures show results for blocks (three full lines, one for each sector) and the PDFs for ridges are also included (as dashed lines) when their impact on ozone is relevant. Adding an additional line for the no block/ridge cases within each sector would complicate the figures substantially.

Another option could be to draw curves for data considering all days without blocks or ridges in any of the three sectors. However, we would be excluding at the same time data from many days with synoptic situations (i.e. blocks or ridges in different sectors) which are associated with very different regional ozone responses. We do not think the information provided by such data would be of better use than the present climatology.

Therefore we have preferred keeping these figures simple and have left them as they were. Note, however, that we have added some additional dashed lines for the ridge cases because their impact on summer O3 is stronger than previously found after using the improved catalogue.

3) Sections 3 and 4 provide a very comprehensive overview about the effects of blocks and ridges on seasonal MDA8-O3. To allow easier comparison among individual cases. I suggest combining Figures 7 and 13 in one 4-panel illustration.

All Figures related to extremes (i.e. Figures 7 and 13 of the ACPD manuscript) have now been combined into a single Figure 7 with four panels. Note, however, that we have removed the panel for ATL ridges in winter and added a panel for EUR ridges in summer instead. This is because, after using the improved catalogue of subtropical ridges, their impact during summer/winter is stronger/weaker than previously thought. The figure for ATL ridges in winter has now been moved to the supplement.

Anonymous Referee #2

Ordóñez et al. present an analysis of the impacts of high-latitude blocks and sub-tropical ridges on surface air quality, particularly ozone, in Europe. The authors use a previously published catalog of these meteorological features to study spatial and temporal changes in ozone concentrations. The connections are made clear and the statistical analysis is sound. The manuscript is clear and pretty concise; in general it is very well-written. I believe the manuscript is high quality and requires limited revision. I recommend publication of this manuscript in ACP after the authors' address the following minor comments.

General Comments

- The most significant issue with this manuscript is the lack of a mechanistic discussion and/or a clear explanation detailing the significance of the results. I disagree that the results are directly usable by modeling communities in assessing changes and validating CCMs. Much more quantitative metrics are necessary. An attempt is made to broaden the results in the final section, but it is not overly convincing. Indeed, many of the relationships presented are not terribly surprising; the relationship between ridging/blocks and air pollution has been known for decades and even presented within this manuscript as a given. What is the significance of these results? Can a predictive

model be developed? Can the results be used to better understand changes brought about by climate change? Can the seasonal frequency or intensity of the EUR blocks be used to summarize seasonal ozone statistics? How can all of this be related to meeting the ozone air quality target? These are the metrics that would make this work more useful to policymakers and model developers alike.

First of all, towards the end of the last section of the ACPD manuscript we wrote: "our results provide quantitative assessments of the response of surface ozone to circulation changes which can be easily exploited to evaluate CTMs". We have kept that sentence but have removed the word "easily" as we are aware that evaluating CCMs or CTMs is not trivial.

Nevertheless, we still think that some of our results are directly applicable to the evaluation of models. The composite maps showing the average responses of seasonal O3 to blocks and ridges within specific sectors, the results on the impact of these synoptic patterns on the occurrence of high O3 extremes, as well as the changes in the PDFs of O3 reported here can be checked for both models and observations. This would provide quantitative comparisons between modelled and observed ozone responses to circulation changes, which may be relevant if atmospheric circulation does not remain stationary in a changing climate. However, models have biases and quite often do not reproduce the observed PDF of ozone, which would complicate some of these evaluations.

Despite the comments above, we completely understand the reviewer's concerns and have tried to address some of her/his questions in this regard. In order to do so, we have built multiple linear models to predict some seasonal ozone statistics as a function of the frequency of occurrence of blocks and ridges within specific sectors. After some sensitivity tests, we have decided to show results from a model predicting the interannual variability of the number of exceedances of the local 90th percentile of O3 in summer (Eq. 1) and another model predicting the winter mean O3 mixing ratios (Eq. 2). We will omit the results from analogous models of the number of exceedances

of the European air quality target (∼60 ppb MDA8 O3), as this is based on a predefined threshold which is not often exceeded at some locations. Because of this we will not attempt to relate our results to meeting the ozone air quality target as suggested by the reviewer.

We have included a new section (before Summary and Discussion) where we describe these statistical models (Eq. 1 and Eq. 2) and discuss the results. Maps of the co-efficients of determination ($R^2$) from the two models are displayed in the new Figure 13. The results prove that the frequency of blocks and ridges can explain a relatively large fraction of the interannual variability of O3 over some areas. The significance of the results is discussed by comparing them with those of previous work which tried to assess the role of the atmospheric circulation in the interannual variability of near-surface ozone (see reply to next comment). Finally, we have also added a couple of sentences to the Summary and discussion as well as to the last section to reinforce the significance of the results.

- Regarding the above comment, the authors provide a nice summary of prior meteorology-ozone research in the introduction. It would be very instructive to revisit those studies later in the manuscript since many of the previous connections can be recast or mechanistically explained using these prior studies.

Following the previous comment, we have put the results from the linear models (Eq. 1 and Eq. 2) in the context of previous analyses which explored the possibility of pre-dicting the interannual variability in the ozone concentrations by using linear models on one or more meteorological variables. First, we have compared our results with those of Pausata et al. (2012), who reported correlations of monthly ozone in Eu-rope with the North Atlantic Oscillation index (NAOI) in winter and a modified NAOI in summer. We have also compared our results with others for the east of the US: the correlations found for the number of summer ozone pollution days with the occurrence of mid-latitude cyclones (Leibensperger et al., 2008), the relationship between surface ozone variability and the jet position projected by Barnes and Fiore (2013), and the

variance explained by a linear model of summer mean ozone on some meteorological fields representing the polar jet frequency and the position of the Bermuda High west edge (Shen et al., 2015). We think that this is instructive enough while keeping the manuscript as concise as possible.

- Why is the spatial aggregation of ozone data (the Schnell data) of better use than observations from individual sites? It seems that the averaging of the observations will mask out some variability and localized features, which may provide valuable mechanistic insights. This is also the case when discussing changes in the PDFs.

We do not intend to suggest that the spatial aggregation of Schnell is of better use than observations from individual sites. However, it is very appropriate for the analyses conducted in this manuscript. It is a very comprehensive compilation that combines MDA8 O3 data from two different networks (EMEP and Airbase) and excludes traffic sites, where titration by NO and other local effects may lead to strongly localized features that are far from the focus of this work. In the introduction (Sect 1) we clearly indicate that this study is focused on the role of the synoptic scale, so the horizontal resolution (1°x 1°) of this gridded dataset is fit for purpose.

Moreover, even when we admit that the use of gridded air quality (AQ) datasets is not essential for these types of analyses, it is a relatively common practice in studies which are not particularly focused on localized features as it is our case. As an example, this dataset has already been used by Otero et al. (2016) for the analysis of the impact of the atmospheric circulation and meteorological conditions on surface ozone in Europe. A number of studies have also averaged O3 (e.g. Leibensperger et al., 2008; Shen et al., 2015) or PM2.5 data (e.g. Tai et al., 2010, 2012) over relatively large grid cells, while other analyses have even used satellite observations of aerosol optical depth at a similar resolution as here (e.g. Jia et al., 2015), to examine relationships of AQ with synoptic conditions and meteorological parameters.

- The terminology used in the manuscript is generally spot-on, but I question the use

of the terms "ridge" and "blocks". Blocks usually imply a long lasting atmospheric pattern (indeed, a ridge) that actually "block" and divert the normal flow of extratropical cyclones. The term block, in my experience, has a lot less to do with latitude (as used in this manuscript), but rather the persistence. The terminology of ridge vs. block does become cumbersome as the reader progresses.

First of all, it is important to keep in mind that there is no universally accepted definition of blocking, even though a number of features (some of them mentioned by the reviewer) are widely accepted in all definitions available in the literature. This has resulted in a number of blocking indices focusing on different characteristics of blocks (see e.g. review by Barriopedro et al., 2010b). Note that in this particular work we are imposing the flow reversal condition for blocking, and not for ridges, thus making this a main criterion for the clear distinction between them.

We understand that the terminology ridge vs. block may become cumbersome, but this terminology is needed. In the second last paragraph of the introduction we clearly state that "There are conceptual differences in the definition of blocking and subtropical ridge patterns although sometimes they induce similar anomalies in the surface climate". There is indeed some overstatement when associating long lasting anticyclonic systems and their effects such as summer heatwaves to blocking (see e.g. García-Herrera et al., 2010). As a consequence, we see a clear need to distinguish high latitude blocking structures from lower latitude systems, including the extensions of subtropical high pressure systems to the mid latitudes, commonly denominated subtropical ridges. Note that the standard definitions of atmospheric blocking consider high latitude quasi-stationary anticyclones associated with a reversal of the prevailing westerly flow (e.g., Rex, 1950a,b; Treidl et al., 1981; Barriopedro et al., 2006, 2010b). Unlike canonical blocking systems, the low latitude structures do not necessarily fulfil the condition of wave-breaking occurrence (Woollings et al., 2011; Masato et al., 2012; Santos et al., 2013) and their impacts on European surface temperatures are different from those of canonical blocking (Sousa et al., 2016b, and references therein). We

have also proved that their effects on surface air quality can be rather different.

We think we have made a clear distinction between the characteristics of high-latitude blocks and subtropical ridges in the introduction as well as in Section 2.2 (Climatology of high-latitude blocks and ridges). Therefore, we have not modified that text, with the exception that now we make it clear that our blocking detection algorithm requires the flow reversal condition.

- Can one block or ridge migrate from one "box" to another? If so, how does that influence ozone?

As blocks and ridges tend to move eastwards, they can indeed migrate from one sector to another. However, a Lagrangian analysis of how specific blocks or ridges may affect O3 at a given location (or in different areas of the continent which may be exposed to very different chemical regimes) as they move eastwards is out of the scope of this work.

In this study we have mainly focused on the impact on ozone of blocks and ridges with centres over sufficiently large longitudinal sectors ($30°$), so that we have enough data to generalise the interpretation of the results and in some cases even assess the effect of persistence. The examination of the impacts over narrower $15°$ degree longitudinal subsectors (covering the west and east flanks of the ATL, EUR and RUS sectors) indicates that the ozone anomalies tend to move eastwards together with the block and ridge centres. For instance, Figure C1 illustrates how the positive ozone anomalies in summer on days with blocks (left) and ridges (right) move eastwards when their centres are respectively found in the east flank of the ATL sector (ATE, $15°$–$0°$ W, upper panels), the west of the EUR sector (EUW, $0°$–$15°$ E, middle panels) and the east of the EUR sector (EUE, $15°$–$30°$ E, lower panels). Please zoom on the figure to see these features.

In the conclusions we clearly state that "ozone anomalies often follow the longitudinal position of the block and ridge centres". In different parts of the text we also mention

that the impact of ridges on ozone is more sensitive to their specific location than in the case of blocks, although we also acknowledge that the impact of blocks on extremes can be sensitive to their location (see last paragraph of Section 3.1). Taking all this into account, we expect the reader to get an overall picture of how the typical eastward migration of blocks and ridges affects surface ozone.

- A number of negative anomalies, generally in the Iberian Peninsula and Italy. These areas have large values, but go unexplained. I recommend adding some discussion about these areas.

We have examined additional daily fields from the NCEP/NCAR meteorological reanalysis (e.g. total cloud cover, downward shortwave and longwave radiation fluxes at surface, temperature at 2 m, horizontal wind speed at 850 hPa) to better understand the origin of some of the negative ozone anomalies, not only in southern Europe but also in other regions. Such meteorological fields are introduced in Section 2.1, but we do not show composite anomaly maps for them to avoid overloading the main text or the supplement with many figures. We have simply introduced some discussions in different parts of the text to explain how the anomalies in these fields relate to the negative ozone anomalies found for some regions.

Specific Comments

- Page 2, Line 11: The degree of significance used in the confidence intervals is unclear and should be specified.

We indicate now that the range given is for the 5 to 95% confidence interval.

- Page 2, Lines 12-14: The identity of these indirect radiative effects are a mystery to the casual reader. Maybe add a few more descriptive words.

In our reference to Stich et al. (2007) we have very briefly indicated that this indirect radiative forcing is due to ozone effects on plants and the resulting reduction in carbon dioxide ($CO2$) uptake. This occurs because ozone is harmful to vegetation and

also causes stomatal closure. As indicated by Stich et al. (2007), increases in tropospheric ozone concentration affect plant productivity and significantly suppress the global land-carbon sink. In consequence, more carbon dioxide ($CO_2$) accumulates in the atmosphere as tropospheric ozone increases.

- Page 2, Line 18: The more recent review by Fiore et al. [2015] may be a better inclusion. Reference: Fiore A. et al., Air quality and climate connections, J. Air Waste Management, 65 (6), 2015.

Reference added.

- Page 3, Line 3-5: A daily NAO index is generated and might prove useful. I cannot speak for the results in Sousa et al. (2016), but I would not discount the NAO index simply because of a perceived monthly timescale.

The referee is right. We have slightly changed that sentence to avoid any reference to the NAO index. Now we simply stress the importance of the examination of daily circulation indices and the day-to-day variability of specific synoptic patterns versus monthly teleconnection indices, as the latter may hide some features at the intra-monthly time scale.

- Page 5, Lines 1-4: a) How trustworthy is the NCEP/NCAR reanalysis I compared to more recent reanalysis projects? Why not use ECMWF-Interim? Or MERRA? Or JRA? All of these have higher spatial resolution and may provide a better picture of the meteorological landscape. b) Also, the placement of the description of met. data here is curious; it could be moved to the beginning of Section 2.2.

a) Please see reply to the first question by referee #1 regarding the choice of the NCEP/NCAR reanalysis dataset for the identification of blocks and ridges.

We are also confident that the NCEP/NCAR reanalysis accurately describes the main flow patterns and meteorological fields analysed here. Previous analyses have already shown that this reanalysis dataset is trustworthy for establishing relationships of AQ

with synoptic patterns and meteorological parameters (e.g. Leibensperger et al., 2008; Tai et al., 2010, 2012; Shen et al., 2015; Chang et al., 2016). Recently, Jia et al. (2015) have also found AQ-synoptic relationships in China which are robust across the NCEP/NCAR and ERA-interim reanalyses. In addition, we are currently working on a manuscript that examines the impact of blocks and ridges on winter PM10 in Europe (Garrido-Perez et al., in prep.). For that particular work we are using the ERA-Interim reanalysis at higher spatial resolution. We confirm that the meteorological fields and flow patterns observed under the presence of blocks and ridges are consistent between the NCEP/NCAR and ERA-Interim reanalysis datasets.

b) Regarding the placement of the description of the meteorological data, a short description is needed in Section 2.1 because it presents Figure 1. Note that the figure displays the seasonal climatology of ozone and two meteorological fields (MSLP and 850 hPa winds) which will be used later to describe the main flow patterns associated with blocks and ridges. We have even extended that paragraph to introduce some additional meteorological fields that we analysed to explain the origin of some negative ozone anomalies (following the recommendation by the same referee).

- Page 5, Lines 9-11: This is curious. Why is this the case? It is interesting that effort is placed into the winter months since ozone levels are so comparatively low in that season, surely lower than in the autumn.

It is obvious that the main concern regarding European surface ozone should be placed on the spring and summer months, when the highest ozone levels are registered. Nonetheless, we also evaluated the impact of blocks and ridges on ozone during the other two seasons and found strong responses in winter. We admit that winter ozone levels are low in Europe, but the results shown in the manuscript are very relevant since they point to a strong impact of some of the synoptic situations analysed here (in particular EUR blocks and to a lesser extent ATL ridges) on the surface levels of other air pollutants in that season. This is hypothesized in Section 5 (Summary and discussion). In fact, we are currently investigating the role of blocks and ridges in surface

PM10 over Europe (Garrido-Perez et al., in prep.), and have found that the strongest signatures can be seen during the winter months and that they are even stronger than those shown for ozone in this manuscript. Therefore, the analysis of the winter ozone data should be kept in the manuscript as it has important implications for present and future work on other pollutants. Note also that, for the reasons indicated above, we have removed some of the figures on the impact of ridges on winter ozone. Our analyses on the impact of ridges are now more focused on the summer season.

Figure C2 shows that, as anticipated by the reviewer, European ozone levels in autumn (SON) are somewhat higher than those in winter over most of Europe (compare with bottom panel of Figure 1 of the main text), but on average still rather low.

We have carefully examined the yearly cycle of ozone over different areas of the continent and have found that, overall, September is the only month in which ozone mixing ratios over large parts of the continent can be substantially higher than those in the other winter and autumn months. As autumn is a transition season between summer and winter, surface ozone over some areas can respond very differently to blocks around the beginning and towards the end of the season. Note that this has not been found for spring. As an example, Figure C3 shows the composite anomalies of ozone on days with EUR blocks during September (top) and November (bottom). The positive anomalies over the north of the continent in September are small but occur over the same regions as in summer (Figure 4 of main text, upper right panel). The negative anomalies in November, in particular over the northwest of Europe, resemble those found for winter (Figure 10 of the main text, middle panel). As a consequence, the composite anomalies calculated considering all months in autumn tend to compensate each other. That is the main reason for the moderate anomalies in autumn.

Following this, we have extended the text around the lines mentioned by the reviewer: "For the sake of brevity, autumn will be omitted from our analyses, as O3 mixing rations are not particularly high in this season and the features that blocks and ridges imprint on surface ozone are weaker than those in the other seasons. The reason for

this is that during autumn there is a strong transition, with ozone responses to these synoptic patterns in September similar but smaller in magnitude than those in summer, while in November such responses resemble those of winter, leading to an overall compensation of positive and negative ozone anomalies".

- Page 5, Line 29-31: Low values of ozone do not necessarily indicate that there is a preferred mechanism (decreased formation, increased loss, or both of ozone). This sentence hints at attribution that is not described.

The reviewer refers to the sentence "The bottom panel of Figure 1 highlights that such processes take place over most of the continent, in particular in regions of elevated emissions such as the Benelux and the Po Basin". To avoid attributing the low ozone levels to any specific mechanisms which we have not evaluated with a model, we have changed this sentence to only indicate what the figure shows. Now it says: "The bottom panel of Figure 1 shows the low winter ozone levels over most of the continent, in particular in regions of elevated emissions such as the Benelux and the Po Basin".

- Page 6, Line 7: With all meteorological-air quality studies it is important to know the sensitivity of the meteorological metric to the assumed parameters. a) How important is this area requirement? b) What about a weaker, but persistent ridge, which may be missed by this algorithm? c) How is the "best" latitudinal break between ridges and blocks determined?

a) First of all, there was an error with the definition of blocks in the original manuscript, where we said: "blocks are detected by applying a simplified version of the method described by Barriopedro et al. (2006) on the NCEP/NCAR reanalysis. By definition, blocks are defined as large-scale departures of Z500 above a seasonally varying threshold and must fulfil some conditions on extension and minimum duration; for this study we have considered that blocks must have a minimum horizontal extension of $2 \cdot 10e6$ km2 during their whole lifetime and a minimum duration of 5 days". We have modified the description, because the blocking detection method by Barriopedro et al.

(2006) is actually based on reversals of the Z500 meridional gradient. We apologize for the misunderstanding. The corrected text says "Blocks are detected by applying a simplified version of the method described by Barriopedro et al. (2006), where they are defined as large-scale reversals of the Z500 meridional gradient and must fulfil some conditions on area overlap during consecutive days and minimum duration. For this study we have considered that reversals must have a minimum longitudinal extension of 12.5° during their whole lifetime and a minimum duration of 5 days".

Therefore, the minimum extension requirement of is not 2·106 km2 but 12.5° in longitude. Nevertheless, the temporal and spatial thresholds of this method were thoroughly tested by Barriopedro et al. (2006) and we are confident that they are quite robust. As mentioned in that paper, previous automated methodologies used blocked region extension criteria ranging from 7.5° to 18.75° but, after testing the algorithm, at least 12.5° were required to ensure that a blocking pattern exists. This is in line with other blocking indices recently used in the literature.

b) Like any other automated detection method, this algorithm may miss some events and even result in some questionable detections. However, from our experience, we are not concerned about the possibility that weak, but persistent ridges, may be missed by the algorithm. Note that weak ridges in the European continent would be hardly detectable from the climatology in seasons like spring and summer. In addition, even if they were easy to detect, they would be associated with small anomalies in the meteorological fields and consequently in the air pollutant concentrations.

c) The movement of this "latitudinal break" according to the season is associated with the northward migration of the jet stream and the mid-latitude circulation from winter to summer. The limits are determined based on previous work and sensitivity analyses. We have added the following sentence to the text: "The optimal latitudinal break relies on previous work about winter climatologies of ridges in the eastern Atlantic (Santos et al., 2009b) and subsequent sensitivity analyses to calibrate their seasonal frequencies in the area of study (Sousa et al., 2106b)".

- Page 6, Line 7: How long does the average block persist? This is important information for the later analysis of the buildup of ozone during an episode.

The duration condition (at least 5 days) imposed in the detection method is common to many blocking studies. It is also well-known that the average blocking duration is ~8-10 days, longer in winter than in summer. However, the average persistence of a block is not the most relevant information for this study. Note that the residence time of a block in a given 15° or 30° longitudinal box could be lower than that, because blocks (and even more blocking centres) tend to move with time, even when they are considered to be quasi-stationary. A very persistent block (e.g. > 15 days) could have a relatively moderate effect on ozone if it is not stationary.

As a consequence, what we try to address in this analysis (by defining longitudinal sectors) is how the location of blocking centres and to some extent their residence times impact the build-up of ozone. This is now mentioned in Section 2.2 shortly after defining the sectors: "Therefore, the use of specific longitudinal sectors is useful to assess how the location of blocking/ridge centres as well as their residence times impact ozone over different areas of the European continent." Some of our subsequent analyses in the manuscript provide evidence that the persistence of blocks over a given sector is indeed relevant for the build-up of ozone (see e.g. Figures 4 and 6 for EUR blocks in spring and summer).

- Page 6, Line 12: What happens if criteria ii is not met? Is it considered a block?

Blocks and ridges are detected through separate algorithms. Both types of patterns tend to be mutually exclusive, but failure to meet criterion ii does not involve the detection of a block in that sector. This is explained below.

First of all, as already mentioned, we have corrected the description of the blocking detection algorithm at the beginning of Section 2.2. Now we make it clear that the detection of blocks (based on Barriopedro et al., 2006) requires the presence of large-scale reversals of the meridional Z500 gradient, while ridges are detected with a different

method based on Z500 thresholds. As indicated in the revised version of the text, "a ridge is detected in a south box if (i) at least 75 % of the grid points are above the 80th percentile of the specific 31-day moving average Z500 climatology during the period 1950-2012 and (ii) no more than 50 % of the grid points of the north box are above the same threshold". Both conditions are imposed to ensure that subtropical ridges are detected as low to mid latitude anomalies of the Z500 field.

Failure to meet condition ii does not involve the detection of a block, because this would still require the presence of a reversal in the meridional Z500 gradient. In addition, as Z500 anomalies tend to be negative to the south when high latitude blocks are detected (see e.g. left panels of Figure 3 for EUR blocks as well as left panels of Supplementary Figure S2 for ATL blocks), it is unlikely that a block is detected over a given sector if condition (i) has already been met.

- Page 6, Lines 23-25: What is the variability in the frequency of blocks and ridges?

As we indicate in that part of the text now, the interannual variability can be quite large. We show it for each sector and season in the new Supplementary Figure S1.

- Page 7, Line 23: Isn't the requirement in Section 2.2 that the block must last at least 5 days? If the system is moving so rapidly through the region (< 3 days), is it really a "block"?

Yes, blocks must last at least 5 days. We have also indicated above that blocks can change sector, but this part of the text does not involve that they move so rapidly through the region. The sentence said: "If in the calculation of the composites we only consider days when blocking centres have stayed over the EUR sector for at least 3 days, the extension of the regions with anomalies higher than 5 ppb increases and there are larger areas where they exceed 10 ppb (Figure 4, bottom panels)". This simply means that if we exclude data from the first two days a block is in the sector the ozone anomalies become higher, indicating the effect that the persistence of blocks has on the build-up of ozone.

We understand that the sentence is confusing and have changed "we only consider days when blocking centres have stayed over the EUR sector for at least 3 days" to "we only consider data from the third day blocking centres have stayed over the EUR sector". We have also corrected the caption of Figure 4 to avoid this misunderstanding.

- Page 8, Line 15: The Scandinavian PDF with EUR blocks looks very similar to the climatology, perhaps even more so than Iberia. Is it really significantly different?

Lines 14-15 on page 8 of the ACPD manuscript said "The results from the two-sample K-S test reveal that the O3 distribution on days with EUR blocks differs from that of the seasonal climatology (at the 0.1% level) for all regions except Iberia in spring". Here we are comparing the distributions given by red lines (EUR blocking) and grey bars (climatologies) for all regions in spring (Figure 5, upper panel). Note that the "NE Scandinavia" region has the narrowest of all probability density functions (PDFs). Because of this it is very difficult by just visual inspection of the PDFs to assess whether they might differ more in this region than in others.

We have double-checked the results from the two-sample Kolmogorov Smirnov-test. This is a non-parametric test used to assess whether two data samples come from the same distribution. This is done by looking for the largest differences (in absolute value) between their cumulative distribution functions (CDFs) and comparing them with a given critical value. Please see Figure C4 with the CDFs of all data (black) and EUR blocks (red) for Iberia (left) and Scandinavia (right) in spring. It is still difficult to detect visually, but the maximum differences between the CDFs are a bit larger for Scandinavia. Moreover, the CDFs start to differ later (at around 40 ppb) for Iberia than for Scandinavia. Results from the test confirm that our previous statement is right: the two data samples are statistically different at the 0.1 % level for Scandinavia and the other regions but not for Iberia.

Therefore, we have not modified the text. Note that in the ACPD manuscript we simply mentioned what the test detects, but we did not comment further on the Scandinavian

case, because the impact of EUR blocks during spring is larger over other areas. We also prefer showing the PDFs instead of the CDFs in the main text as the first ones are easier to visualise in most cases.

- Page 8, Line 20: Why is this position of the block so efficient at increasing ozone in UK/N. France?

We have compared maps with anomalies for a number of meteorological fields and added this sentence to the revised version: "This synoptic situation . . . is efficient at increasing ozone in the British Isles and the northwest of France, because it is associated with a decrease in cloud cover and therefore an increase in the downward shortwave radiation flux, as well as increases in daily maximum temperature and decreases in 850 hPa wind speed over those regions (not shown)".

- Page 10, 1st paragraph of section 4. I think that this needs a bit more thought and discussion. The reason for anomalously low ozone values that is provided suggests the cut off of marine air that brings in background ozone. Yet, Figure 10 shows that locations susceptible to marine air (as indicated by arrows) have negative anomalies, particularly in the RUW case (bottom of Fig. 10).

As mentioned above, we have analysed additional meteorological fields from the NCEP/NCAR reanalysis to discuss the origin of some of the negative ozone anomalies. We have extended this part of the text to indicate the fields associated with the negative ozone anomalies on days with EUR blocks (middle panel of Figure 10), while the fields related to the largest negative ozone anomalies on days with RUW blocking (bottom panel of Figure 10) are mentioned below. Please see reply to next question.

- Page 11, Line 11-13: As mentioned above, this is not what I am seeing. I agree that continental air is advected northward, but the largest anomalies appear to be receiving westerly wind from the Atlantic.

This is a continuation of the previous question and here is where we have explained

the origin of the largest negative ozone anomalies seen in the bottom panel of Figure 10. Note, however, that we do not completely agree with the referee. The regions with westerly advection (e.g. the British Isles and the Iberian Peninsula) have moderate ozone anomalies, while the largest negative anomalies are seen for a large region around Germany and a small region in northern Scandinavia. Now the reasons for such high anomalies are explained in this part of the text.

- The contour lines within the figures containing maps are very hard to distinguish. I recommend thickening them up before publication.

Done it for all the contour lines representing MSLP in the ozone anomaly maps. Not done for the contour lines in the Z500 maps (Figure 3 and two Supplementary Figures) as the contours there are easy to distinguish and we already used a thick line to highlight the position of the 560 dam isohypse.

References (only those which were not present in the ACPD manuscript are listed here)

Barnes, E. A. and Fiore, A. M.: Surface ozone variability and the jet position: Implications for projecting future air quality, Geophys. Res. Lett., 40, 2839–2844, doi:10.1002/grl.50411, 2013.

Barnes, E. A., E. Dunn-Sigouin, G. Masato, and T. Woollings: Exploring recent trends in Northern Hemisphere blocking, Geophys. Res. Lett., 41, doi:10.1002/2013GL058745, 2014.

Chang, L., Xu, J., Tie, X., and Wu, J.: Impact of the 2015 El Nino event on winter air quality in China, Nature Scientific Reports, DOI: 10.1038/srep34275, 2016.

Davini, P., C. Cagnazzo, S. Gualdi, and A. Navarra: Bidimensional Diagnostics, Variability, and Trends of Northern Hemisphere Blocking, J. Clim, 25, 6496-6509, 2012.

Garrido-Perez, J. M., C. Ordóñez, et al.: Strong signatures of high-latitude blocks and subtropical ridges in winter PM10 in Europe, in prep.

Jia, B., Wang, Y., Yao, Y., and Xie, Y.: A new indicator on the impact of large-scale circulation on wintertime particulate matter pollution over China, Atmos. Chem. Phys., 15, 11919-11929, doi:10.5194/acp-15-11919-2015, 2015.

Leibensperger, E. M., Mickley, L. J., and Jacob, D. J.: Sensitivity of US air quality to mid-latitude cyclone frequency and implications of 1980–2006 climate change, Atmos. Chem. Phys., 8, 7075-7086, doi:10.5194/acp-8-7075-2008, 2008.

Rex, D.F.: Blocking action in the middle troposphere and its effect upon regional climate, Part I: an aerological study of blocking action, Tellus, 2, 196–211, 1950a.

Rex, D.F.: Blocking action in the middle troposphere and its effect upon regional climate, Part II: the climatology of blocking action, Tellus, 2, 275–301, 1950b.

Santos, J. A., Woollings, T., and Pinto, J. G.: Are the Winters 2010 and 2012 Archetypes Exhibiting Extreme Opposite Behavior of the North Atlantic Jet Stream? Mon. Weath. Rev., 141, 3626-3640, doi:10.1175/MWR-D-13-00024.1, 2013.

Shen, L., L. J. Mickley, and A. P. K. Tai: Influence of synoptic patterns on surface ozone variability over the eastern United States from 1980 to 2012. Atmos. Chem. Phys., 15, 10925–10938, doi:10.5194/acp-15-10925-2015, 2015.

Tai, A. P. K., L. J. Mickley, D. J. Jacob: Correlations between fine particulate matter (PM2.5) and meteorological variables in the United States: Implications for the sensitivity of PM2.5 to climate change, Atmospheric Environment, 44, 3976-3984, 2010.

Tai, A. P. K., Mickley, L. J., Jacob, D. J., Leibensperger, E. M., Zhang, L., Fisher, J. A., and Pye, H. O. T.: Meteorological modes of variability for fine particulate matter (PM2.5) air quality in the United States: implications for PM2.5 sensitivity to climate change, Atmos. Chem. Phys., 12, 3131-3145, doi:10.5194/acp-12-3131-2012, 2012.
Treidl, R. A., Birch, E. C., and Sajecki, P.: Blocking action in the northern hemisphere: a climatological study, Atmos. Ocean., 19, 1–23. doi:10.1080/07055900.1981.9649096, 1981.

(3) Authors' changes to the manuscript

The revised versions of the manuscript and supplement, with all changes highlighted in blue, will be uploaded.

[Figure]

[Figure]

**Fig. 1.** Figure C1

[Figure]

[Figure]

[Figure]

---

## Author Comment (AC2) · 24 Jan 2017

Please see replies to both referees in a single file which has been uploaded as an author comment.

---

## Author Comment (AC3) · 24 Jan 2017

Please see replies to both referees in a single file which has been uploaded as an author comment.

---

## Author Response (AR1)

**Dear editor,**

**Please see our point-to-point response to the reviews, which has been uploaded as a single Author Comment (AC1).**
**The pdf file can be downloaded from http://www.atmos-chem-phys-discuss.net/acp-2016-832/#discussion (see AC1:**
**"Replies to referees' comments"). It contains our replies to their questions and, if appropriate, information on the**
**relevant changes introduced in the revised version the manuscript.**

**The following pages of this response file also include the revised version of the manuscript with all changes**
**highlighted in blue colour.**

**Best regards,**

**Carlos Ordóñez and co-authors**

[revised manuscript text omitted]